# The exonic SNP rs11676272-C risk allele mediates diet-induced obesity and reduces enhancer activation

Weina Wang[1,2], Yue Li[1], Sheng Dong [1], Yuwei Liu [1], Chenghang Guo[1], Yuzhe Su [1], Wei Tian [1], Xiaoyu Hu [1,3,4✉] & Zhenshan Wang [1,3,4,5✉]

## Abstract

**Genome-wide association studies (GWASs) have identified hundreds of obesity-associated SNPs, but establishing their causality remains challenging. Here, we demonstrate that rs11676272, located in the ADCY3 gene, is a functional causal variant for obesity susceptibility. Bioinformatic analyses and dual-luciferase reporter assays indicate that the rs11676272 region may act as a human-gained enhancer regulating ADCY3 expression. In HEK293T cells, CRISPR-Cas9-mediated single-nucleotide editing of rs11676272 (T > C) reduces ADCY3 expression. Moreover, the rs11676272-T allele is preferentially bound by the transcription factor E2F3 to upregulate ADCY3 expression, whereas the rs11676272-C risk allele loses this binding. In vivo, the rs11676272 T > C variant in human ADCY3 (hADCY3) knock-in mice accelerates weight gain under high-fat diet conditions and shortens primary cilia in the ventromedial hypothalamus (VMH). CRISPRa-mediated activation of the hADCY3 promoter region rescues ciliary length in both the VMH and hypothalamic arcuate nucleus of Mut-hADCY3 mice. Our data reveal a causal role for rs11676272 in obesity, offering insight into potential therapeutic strategies.**

**Subject Categories** Chromatin, Transcription & Genomics; Metabolism

## Introduction

Obesity is a metabolic disorder characterized by the accumulation of excess fat in the body, which increases the risk of many diseases, including type 2 diabetes, hypertension, cardiovascular disease and certain types of cancer (Chong et al, 2023; Neeland et al, 2019; Sung et al, 2019). To date, hundreds of single-nucleotide polymorphism (SNP) loci related to body mass index (BMI) have been identified via genome-wide association studies (GWASs) (Rask-Andersen

et al, 2023; Venkatesh et al, 2024). However, the biological functions of most obesity-associated SNPs have not yet been characterized (Šimon et al, 2024).

Adenylyl cyclase 3 (ADCY3) is a transmembrane protein that catalyzes the synthesis of cyclic adenosine monophosphate (cAMP), a second messenger in multiple signaling pathways, particularly neuronal primary cilium signaling. In mice, Adcy3 deficiency results in severe obesity and insulin resistance (Wang et al, 2009). In addition, ADCY3 loss-of-function variants increase the risk of obesity and type 2 diabetes in human cohorts (Grarup et al, 2018). Through GWAS analyses, dozens of SNPs residing in the *ADCY3* gene have been identified as potential contributors to obesity susceptibility in humans (Fitzpatrick and Solberg Woods, 2024; Hammond et al, 2021; Nordman et al, 2008; Stergiakouli et al, 2014; Wang et al, 2010; Wen et al, 2012). Repeated studies on human samples have demonstrated a strong association between rs11676272 (hg38: 24,918,669) and obesity (Stergiakouli et al, 2014; Wang et al, 2010; Warrington et al, 2015). A variant at the rs11676272 locus leads to a serine-to-proline substitution at position 107 in the protein, which may result in reduced enzyme activity (Wen et al, 2012). In a large-scale cohort study involving 50,000 children, the rs11676272 locus demonstrated the most statistically significant association among 20 candidate loci screened for their significant relevance to obesity (Felix et al, 2016). While studies have provided evidence of an association between the rs11676272 T > C variant and obesity, whether rs11676272 is a genuine causal variant and the molecular mechanisms by which it regulates obesity remain unclear.

SNPs can be classified into noncoding SNPs and coding SNPs depending on their location within the genome. Noncoding SNPs typically reside in gene regulatory elements, such as enhancers, which modulate gene expression through genomic interactions (Yang et al, 2022). Chromatin at active enhancer regions has been reported to exhibit histone modifications, including H3K27ac and H3K4me1, which are marks of enhancer regions and are associated with active regulatory elements (Tuvikene et al, 2021). In contrast, coding SNPs have been demonstrated to significantly affect the expression, function, or activity of encoded proteins. For example, missense variants of peripheral myelin protein-22 have been

[1]School of Life Sciences, Institute of Life Science and Green Development, Hebei University, Baoding 071002, China. [2]The First Affiliated Hospital & Liangzhu Laboratory, Zhejiang University School of Medicine, Hangzhou 310058, China. [3]Hebei Innovation Center for Bioengineering and Biotechnology, Hebei University, Baoding 071002, China. [4]Hebei Basic Science Center for Biotic Interaction, Hebei University, Baoding 071002, China. [5]Key Laboratory of Medicinal Chemistry and Molecular Diagnosis, Ministry of Education, Hebei University, Baoding 071002, China. ✉E-mail: Xiaoyu.Hu@hbu.edu.cn; zswang@hbu.edu.cn

reported to disrupt protein conformation and intracellular trafficking (Li, 2015).

Furthermore, specific exonic DNA, beyond its coding function, can regulate the expression of its host genes or nearby genes (Ahituv, 2016). Approximately 15% of human codons possess binding sites for transcription factors (TFs) (Birnbaum et al, 2012; Stergachis et al, 2013), which may function as exonic enhancers (eExons) (Birnbaum et al, 2014). eExons have been widely identified in the genome, and functional studies in mice and zebrafish have revealed that more than half of eExons are functional enhancers of nearby genes (Birnbaum et al, 2012; Ritter et al, 2012). For example, rs13266634 is located within the eExon of the zinc transporter *SLC30A8* gene, and variants of this locus increase enhancer activity in this region, which subsequently increases the risk of type 2 diabetes (Eufrásio et al, 2020). Therefore, the regulatory effect of variants located in eExon region on gene activity cannot be ignored.

Here, we performed bioinformatics analyses, including a meta-analysis of BMI from the UK Biobank and the Genetic Investigation of ANthropometric Traits (GIANT) consortium, RegulomeDB, linkage disequilibrium (LD) analysis and chromatin immunoprecipitation sequencing (ChIP-seq), to identify a functional variant within the *ADCY3* gene for further experimental validation. We demonstrated that rs11676272 acted as an enhancer to regulate ADCY3 expression. Furthermore, we employed a dual-fluorescence reporter assay and CRISPR activation (CRISPRa)-, CRISPR interference (CRISPRi)-, and CRISPR-Cas9-based experiments to validate the function of rs11676272 in HEK293T cells. Additionally, we confirmed that Mut-hADCY3 mice, which carry the T > C variant in hADCY3 knock-in mice, developed obesity when fed a high-fat diet (HFD). Stereotactic delivery of CRISPRa-adeno-associated virus (CRISPRa-AAV) into the ventromedial hypothalamus (VMH) and hypothalamic arcuate nucleus (ARC) of hADCY3 and Mut-hADCY3 mice revealed that activation of the hADCY3 promoter rescued the shortened cilia phenotype observed in Mut-hADCY3 mice. Collectively, these findings illustrate the biological mechanisms by which the rs11676272-C risk variant influences obesity risk in humans.

## Results

### The rs11676272 locus is a potential functional SNP reducing ADCY3 expression

Dozens of obesity-associated SNP loci have been identified in the *ADCY3* gene in multiple human sample cohort analyses (Domingue et al, 2014; Hammond et al, 2021; Nordman et al, 2008; Wang et al, 2010). To investigate their roles in obesity pathogenesis, we extracted SNP loci spanning the entire *ADCY3* gene through a meta-analysis of BMI in the UK Biobank and GIANT databases (Dataset EV1). We identified 123 obesity susceptibility SNP loci associated with the *ADCY3* gene (Fig. 1A). To assess their functional significance, we utilized the RegulomeDB database (Dong et al, 2023), which facilitates annotation and prioritization of human genomic variants, enabling efficient screening of potential functional sites from large-scale genetic variation data. The analysis revealed that rs11676272, located in exon 1 of *ADCY3*,

was the lead SNP associated with obesity ($p = 5.015 \times 10^{-88}$) in a study involving 806,834 samples (Fig. 1A,B; Dataset EV1).

To assess whether the BMI association at the *ADCY3* locus is driven by rs11676272 independently of other linked variants, we analyzed pairwise LD among 123 SNPs using LDlink (Machiela and Chanock, 2015), with the European population from the 1000 Genomes Project Phase 3 as the reference panel. Statistical fine-mapping was performed using the approximate Bayes factor (ABF) framework (Wakefield, 2009) to calculate posterior inclusion probabilities (PIP), assuming a prior variance of 0.04. The 95% credible set was defined as the minimal set of variants with a cumulative PIP exceeding 0.95 (Dataset EV2). Fine-mapping provided strong evidence that rs11676272 is the causal variant underlying the BMI association at the *ADCY3* locus. This variant showed a PIP greater than 0.999, and the 95% credible set contained rs11676272 alone (Fig. 1C; Dataset EV2). Among the 123 variants analyzed, 120 were in LD with rs11676272, including 20 high-LD SNPs ($r^2 > 0.8$), 39 moderate-LD SNPs ($0.5 \leq r^2 < 0.8$), 30 lower-LD SNPs ($0.2 \leq r^2 < 0.5$), and 31 low-LD SNPs ($r^2 < 0.2$). Association strength declined monotonically with decreasing LD, with mean $-\log_{10}P$ values of 76.7 ($r^2 = 0.8–1.0$), 72.2 ($r^2 = 0.6–0.8$), 35.3 ($r^2 = 0.4–0.6$), and 8.8 ($r^2 < 0.2$). This pattern is consistent with an LD-driven signal decay rather than multiple independent effects. Although 14 low-LD SNPs ($r^2 < 0.2$) reached genome-wide significance ($P < 5 \times 10^{-8}$), their association signals were 10–15 orders of magnitude weaker than that of rs11676272 (Dataset EV2), suggesting that they reflect long-range LD or haplotypic effects rather than independent causal variants.

While LD limits causal inference from association data alone, the convergent evidence from fine-mapping ($P = 5.01 \times 10^{-88}$, PIP >0.999, single-variant 95% credible set), functional annotation (exonic location, eQTL evidence, RegulomeDB score of 1b), and experimental validation supports rs11676272 as the causal variant driving the BMI association at the *ADCY3* locus.

Additionally, bioinformatic analyses via the likelihood ratio test (LRT) and combined annotation-dependent depletion (CADD) tools (Schubach et al, 2024) revealed that the rs11676272 (raw score = 1.471778, PHRED = 14.02) risk variant is potentially deleterious and may impair ADCY3 function (Fig. 1D). Based on these findings, we prioritized rs11676272 as a functional variant for further experimental validation.

To investigate whether the rs11676272 variant affects the catalytic function of ADCY3, we sought to construct an isogenic cell line carrying a single-nucleotide substitution. Adenine base editors (ABEs) (Gaudelli et al, 2017) and cytosine base editors (CBEs) (Komor et al, 2016) were unsuitable due to the absence of an appropriate editing window surrounding the rs11676272 locus. Consequently, we employed the CRISPR-Cas9 system coupled with homologous recombination to introduce the homozygous C/C allele at the rs11676272 locus in HEK293T cells. Following cotransfection of sgRNA and homologous recombination templates for 72 h, monoclonal cells were isolated through flow cytometry sorting and limiting dilution assays. Sanger sequencing confirmed successful editing in 1 of 641 clones (Fig. 1E,F), and the predicted off-target site sequence revealed no off-target editing.

Enzymatic assays revealed no significant difference in ADCY3 enzymatic activity between T/T (rs11676272–T allele) and C/C (rs11676272–C allele) cells (Fig. EV1A). Furthermore, structural prediction using the AlphaFold Server yielded a root-mean-square

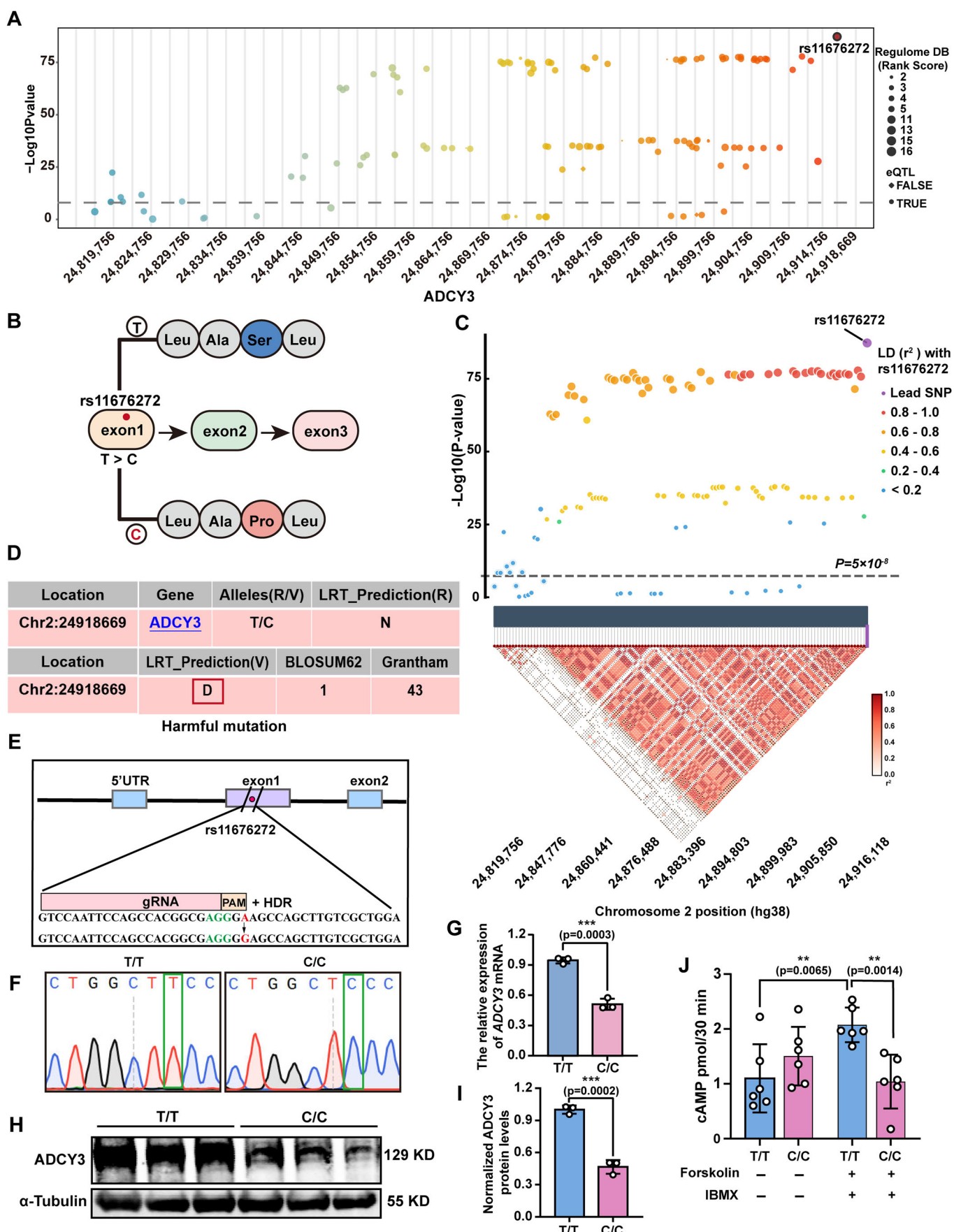

◀    **Figure 1.   Prioritizing rs11676272 locus as a potential functional SNP.**

(A) Plot of obesity-associated SNPs in *ADCY3* SNP locus revealed by GWAS database. Circles represent eQTL loci, while boxes represent non eQTL loci. The size of circles and boxes represent regulome DB rank scores. Dotted line indicates $p$ value $= 1 \times 10^{-8}$. (B) Schematic diagram of amino acid substitution in the rs11676272 locus. (C) Regional association plot and LD structure at the *ADCY3* locus, highlighting rs11676272. The supper panel displays $-\log_{10}(P)$ values for SNP associations with BMI across the *ADCY3* locus on chromosome 2 (hg38). Each point represents a variant, colored according to its pairwise LD ($r^2$) with the lead SNP rs11676272, as indicated in the legend. rs11676272 is shown in purple. The horizontal dashed line denotes the genome-wide significance threshold ($P = 5 \times 10^{-8}$). The lower panel displays the local LD structure ($r^2$) among variants in the region, with darker shades indicating stronger LD. Association strength decreases monotonically with declining LD relative to rs11676272. (D) Harmful mutation analysis in the LRT database. D deleterious. N neutral. U unknown. (E) Schematic illustrating the CRISPR-Cas9 mediated mutation of the rs11676272 locus. The red base is the rs11676272 locus, and the green bases are the PAM sequences. (F) Sanger sequencing of the single-base editing of the rs11676272 locus in HEK293T cells. The green box represents the rs11676272 locus. (G) qPCR detecting the *ADCY3* mRNA levels in T/T and C/C cells. $n = 3$ groups from three independent experiments. (H, I). Western blot probing with anti-ADCY3 antibody in cells carrying different alleles of the rs11676272 locus. α-Tubulin was used as a loading control. $n = 3$ groups from three independent experiments. (J) Intracellular cAMP levels in cells carrying different alleles of the rs11676272 locus with or without forskolin and IBMX treatments. $n = 6$ groups from three independent experiments. All data were presented as mean ± SD, **$p < 0.01$, ***$p < 0.001$ by Student's $t$-test. Source data are available online for this figure.

deviation (RMSD) of 1.3678, indicating that the rs11676272 variant does not substantially alter ADCY3 protein structure (Fig. EV1B). However, qPCR and western blot analysis revealed significantly lower *ADCY3* mRNA and protein levels in C/C cells compared to T/T cells (Fig. 1G–I). Additionally, as ADCY3 catalyzes the synthesis of cAMP from ATP (Zhang et al, 2017), we measured intracellular cAMP levels. Cells were treated with forskolin (an ADCY activator) and 3-isobutyl-1-methylxanthine (IBMX, a phosphodiesterase inhibitor) for 30 min (Roger et al, 2011; Saeed et al, 2018). Basal cAMP levels were similar between T/T and C/C cells in the absence of forskolin and IBMX treatment. However, upon forskolin and IBMX treatment, a substantially lower level of cAMP was observed in C/C cells compared to that in T/T cells (Fig. 1J), indicating that the rs11676272-C risk variant is not sensitive to forskolin or IBMX, which stimulate ADCY3 enzymatic activity. Collectively, these results suggest that the rs11676272-C variant downregulates ADCY3 expression while appearing not to alter its intrinsic enzymatic activity or protein structure.

## The rs11676272 region acts as an enhancer to regulate ADCY3 expression

Since ADCY3 functions primarily through its effects on brain and adipose tissue (Khani et al, 2024; Yang et al, 2022), we investigated the functional role of rs11676272 by retrieving datasets of the human brain and adipose tissue from the ENCODE database (Table EV1). Our analyses indicated that the rs11676272 region is highly enriched with enhancer markers such as H3K27ac and H3K4me1 in brain tissue, with enhancer activity further validated via the VARAdb database (Fig. 2A,B). In contrast, histone modification abundance in the rs11676272 region was consistently weaker in adipose tissue than in brain tissue (Fig. EV1C). Moreover, rs11676272 resides within an ATAC-seq peak, indicating chromatin accessibility (Fig. 2C). These functional annotations provide substantial evidence that rs11676272 is located in an enhancer regulatory region.

We then used a dual-luciferase reporter assay to experimentally validate the allele-specific effect of rs11676272 on enhancer activity. We cloned the *ADCY3* promoter (*ADCY3 pro*, 1500 bp upstream and 500 bp downstream of the transcription start site), *ADCY3 pro-T/T* (*ADCY3 pro* with the rs11676272-T nonrisk allele region containing 518 bp), and *ADCY3 pro-C/C* (*ADCY3 pro* with the rs11676272-C risk allele region containing 518 bp) into luciferase

vectors. These constructs were subsequently transfected into SY5Y, HEK293T, 3T3-L1, and LO2 cell lines, and luciferase activity was measured 48 h later via the dual-luciferase reporter system. Compared with the *ADCY3 pro* control, the *ADCY3 pro-T/T* construct significantly increased luciferase activity, whereas the *ADCY3 pro-C/C* construct reduced luciferase activity in SY5Y, HEK293T, and 3T3-L1 cells. No significant differences were observed in LO2 cells (Fig. 2D). Owing to their high transfection and proliferation efficiencies, we selected HEK293T cells for subsequent validation studies.

To verify the effect of rs11676272 on *ADCY3* expression, we examined *ADCY3* expression via CRISPRa and CRISPRi assays. Five single guide RNAs (sgRNA1, sgRNA2, sgRNA-3, sgRNA-4, and sgRNA-5) targeting the rs11676272 region were designed and individually cloned and inserted into the pEASY-U6 vector. These constructs, along with catalytically dead Cas9 fused to VPR (activator) or KRAB (repressor) domains (Liu et al, 2022), were transfected into HEK293T cells. The qPCR results revealed that sgRNA-5 exhibited the highest inhibition and activation efficiencies (Fig. EV2A,B). The qPCR and western blot results confirmed that sgRNA-5, when combined with CRISPRa, significantly upregulated ADCY3 expression (Fig. 2E–G), while its combination with CRISPRi markedly reduced ADCY3 expression (Fig. 2H–J). Taken together, these findings demonstrate that the rs11676272 region functions as an enhancer to regulate ADCY3 expression.

## The rs11676272-C risk allele has a lower binding affinity for the TF E2F3, which reduces the expression of ADCY3

Given that functional SNPs in enhancers typically exert their functions through recruiting TFs (Kawasaki and Fukaya, 2023; Spitz and Furlong, 2012), we performed TF prediction to identify the TFs that bind to the rs11676272 alleles. Our analyses of multiple cell types revealed that E2F3 and E2F6 may bind to the rs11676272-T allele, whereas SP1 and KLF5 showed a preference for the C allele (Figs. 3A,B and EV3A). Next, we tested whether these TFs are involved in the allele-specific effect of rs11676272 on enhancer activity. The coding sequences (CDSs) of *E2F3*, *E2F6*, *SP1*, and *KLF5* were cloned and inserted into the PLV-3HA-MCS-3Flag vector and individually transfected into HEK293T cells along with the luciferase reporter constructs carrying *ADCY3 pro-T/T* or *ADCY3 pro-C/C*. Notably, the luciferase activity was significantly greater when E2F3 and *ADCY3 pro-T/T* were cotransfected than

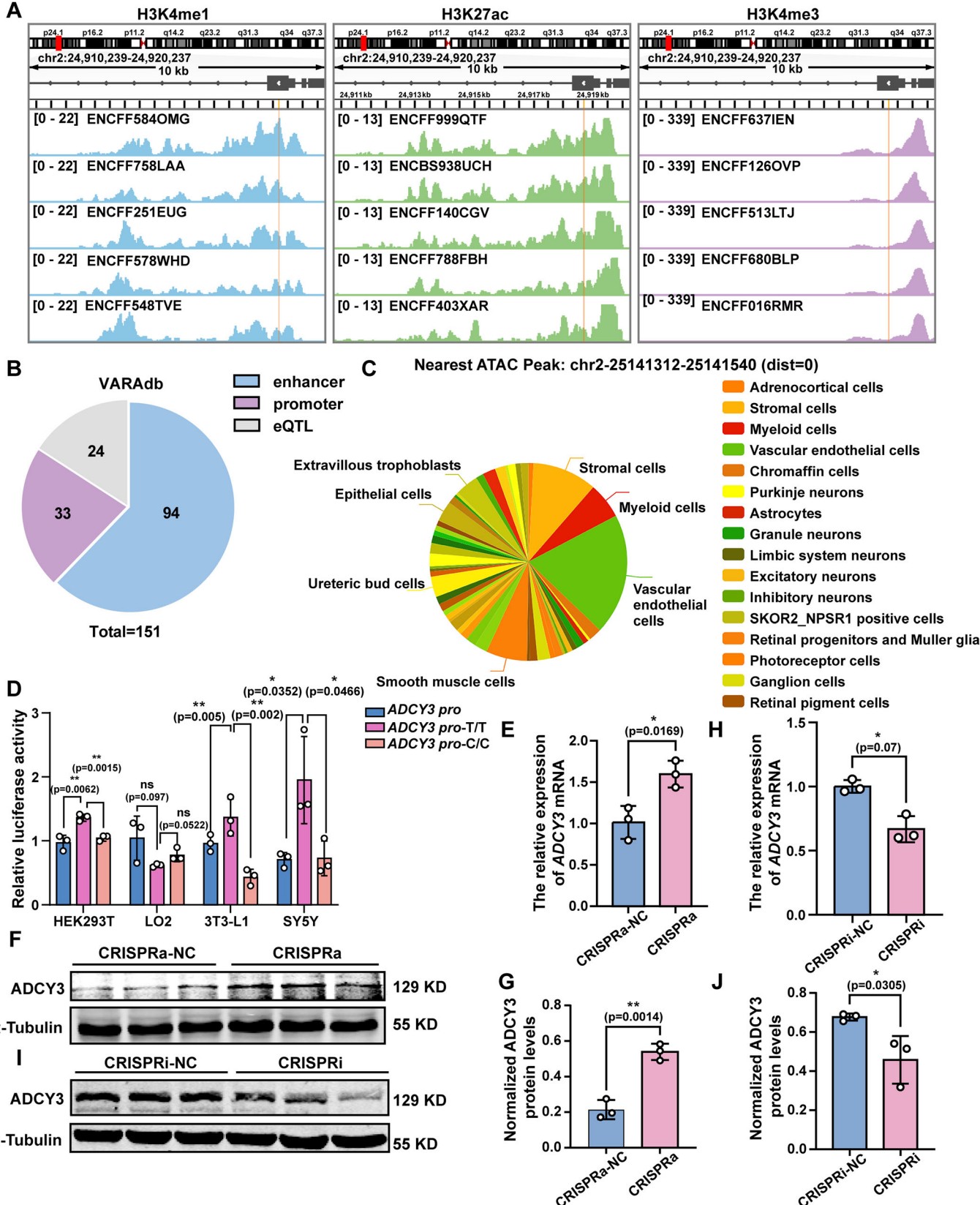

◀ **Figure 2. The rs11676272 variant reduces ADCY3 expression.**

(A) Epigenetic annotations of H3K27ac, H3K4me1, and H3K4me3 marks for the rs11676272 region in brain tissue from the ENCODE database. rs11676272 is indicated by the orange vertical lines. (B) The VARadb database shows that the rs11676272 region exhibits enhancer activity. (C) The 3D SNP database shows the ATAC peak signal surrounding the rs11676272 region in multiple cell lines. (D) Dual-luciferase reporter assays in cells transfected with control vector, *ADCY3* promoter alone, or *ADCY3* promoter containing the rs11676272-T or -C allele. The signals were normalized to firefly signals. $n = 3$ groups from three independent experiments. $p$ values were calculated using one-way ANOVA. (E) qPCR analysis of *ADCY3* mRNA levels in CRISPRa-NC and CRISPRa cells. $n = 3$ groups from three independent experiments. (F, G) Western blot analysis of ADCY3 in lysates from CRISPRa-NC and CRISPRa cells. α-Tubulin was used as a loading control. $n = 3$ groups from three independent experiments. (H) qPCR analysis of *ADCY3* gene expression levels in CRISPRi-NC and CRISPRi cells. $n = 3$ groups from three independent experiments. (I, J) Western blot of ADCY3 in CRISPRi-NC and CRISPRi cells. $n = 3$ groups from three independent experiments. α-Tubulin was used as a loading control. All data were presented as mean ± SD, $*p < 0.05$, $**p < 0.01$. ns no significance, and $p$ values were calculated using Student's $t$-test in E–J. Source data are available online for this figure.

when *ADCY3 pro*-T/T alone was transfected. However, *ADCY3 pro*-C/C showed a significantly lower binding affinity for E2F3 than the *ADCY3 pro*-T/T control did (Fig. 3C). Moreover, there was no difference in luciferase activity between E2F6-overexpressing *ADCY3 pro*-T/T or *ADCY3 pro*-C/C cells and control cells (Fig. 3D). Taken together, these results demonstrate that E2F3 preferentially binds to the rs11676272-T nonrisk allele, thereby increasing *ADCY3* promoter activity. Similarly, we also examined the binding affinity of SP1 and KLF5 for the rs11676272-T and C alleles. The results revealed that *ADCY3* promoter activity was significantly increased following transfection with either SP1 or KLF5. However, the rs11676272 SNP variant did not affect luciferase activity with either SP1 or KLF5 (Fig. EV3B), which may be attributed to the direct binding of SP1 and KLF5 to the *ADCY3* promoter region rather than to the rs11676272 locus.

To validate the binding of E2F3 to different rs11676272 alleles, we analyzed the effects of the rs11676272-T and -C alleles on E2F3 binding via a ChIP quantitative polymerase chain reaction (qPCR) assay. The assay revealed that the rs11676272-T allele but not the C allele could bind to E2F3 (Figs. 3E and EV4A). Similarly, electrophoretic mobility shift assay (EMSA) results confirmed that the T allele probe interacts with E2F3. Furthermore, cold competition experiments demonstrated that the addition of a 200-fold excess of a nonbiotin-labeled T allele probe depleted the biotin-labeled T allele probe from the nucleoprotein complex, whereas the introduction of a 200-fold excess of a nonbiotin-labeled C allele probe maintained a notable degree of binding (Fig. EV4B). These data indicate that E2F3 preferentially binds to the rs11676272-T allele and that the rs11676272-T > C variant significantly reduces the binding affinity of E2F3.

To further assess whether E2F3 binds to the rs11676272 region, we transfected cells with an sgRNA targeting the E2F3 binding site and disrupted the TF–DNA interactions by spatially blocking TF binding via the CRISPRd approach (Shariati et al, 2019). The western blot results revealed that, compared with the control (CRISPRd-NC), the sgRNA targeting the E2F3 binding site (CRISPRd-E2F3) notably reduced ADCY3 expression (Fig. 3F–H). Furthermore, E2F3 overexpression resulted in significant upregulation of ADCY3 expression (Fig. 3I–K), and E2F3 knockdown significantly decreased ADCY3 expression (Fig. 3L–Q), confirming that E2F3 positively regulates ADCY3 expression. With respect to the rs11676272-C allele, compared with that in the control group, the overexpression of E2F3 did not affect the expression of ADCY3 (Fig. EV5A,B). Taken together, these data demonstrate that the rs11676272-C risk allele exhibited a significantly lower binding capacity for E2F3, reducing the expression level of ADCY3.

## Establishment of a mouse model of the rs11676272-C risk variant in hADCY3 knock-in mice

Studies using human cells have provided crucial insights into the functional involvement and potential mechanisms of the rs11676272 variants linked to obesity. However, these in vitro findings cannot fully recapitulate the in vivo situation. Although iPSC-derived neuronal models would facilitate mechanistic studies of this variant, they remain inadequate for modeling complex obesity phenotypes. In contrast, in vivo results provide a more accurate reflection of the physiological condition.

To this end, we conducted a sequence alignment of the *ADCY3* DNA sequences containing the rs11676272 region. Our findings revealed that the *ADCY3* gene sequences within this region exhibit notable divergence between humans and other species (Fig. 4A). A total of 30 species were selected for DNA sequence alignment. The rs11676272 locus predominantly contains the G allele (complementary C allele) in most species, including chimpanzees and two ancient hominid species, Neanderthals and Denisovans, whereas the A allele (complementary T allele) is exclusively present in *Homo sapiens*. This finding suggests that rs11676272 is a derived SNP. Furthermore, a comparative analysis of the human and chimpanzee rs11676272 regions revealed a significant reduction in the enrichment of H3K4me1 and H3K27ac in chimpanzees. No enhancer activity was detected near the rs11676272 locus in mouse or chimpanzee *ADCY3* via ENCODE and European Nucleotide Archive bioinformatics analyses (Fig. 4B and Table EV1).

Moreover, to experimentally verify whether the mouse and chimpanzee rs11676272 homologous regions exhibit enhancer activity, we transfected cells with pGL4.7 dual-luciferase reporter constructs carrying a cytomegalovirus (CMV) promoter alone (CMV) or in combination with exon 1 from humans, chimpanzees or mice containing the rs11676272 homologous region (CMV-human-272T, CMV-human-272C, CMV-chimpanzee and CMV-mouse). The enhancer activity of the chimpanzee rs11676272 homologous region was significantly lower than that of the human rs11676272 region. Additionally, the mouse rs11676272 homologous region exhibited complete loss of enhancer activity (Fig. 4C). These results show that the rs11676272 region has stronger enhancer activity in humans than do homologous regions in chimpanzees and mice. Therefore, investigating the function of the human rs11676272 locus through direct single-base editing in wild-type mice was not feasible.

Since the mouse rs11676272 homologous region lacks enhancer activity, we employed previously established humanized *ADCY3* knock-in (hADCY3) mice (Yang et al, 2022), in which the complete hADCY3 sequence was inserted into the Rosa26 site in the mouse

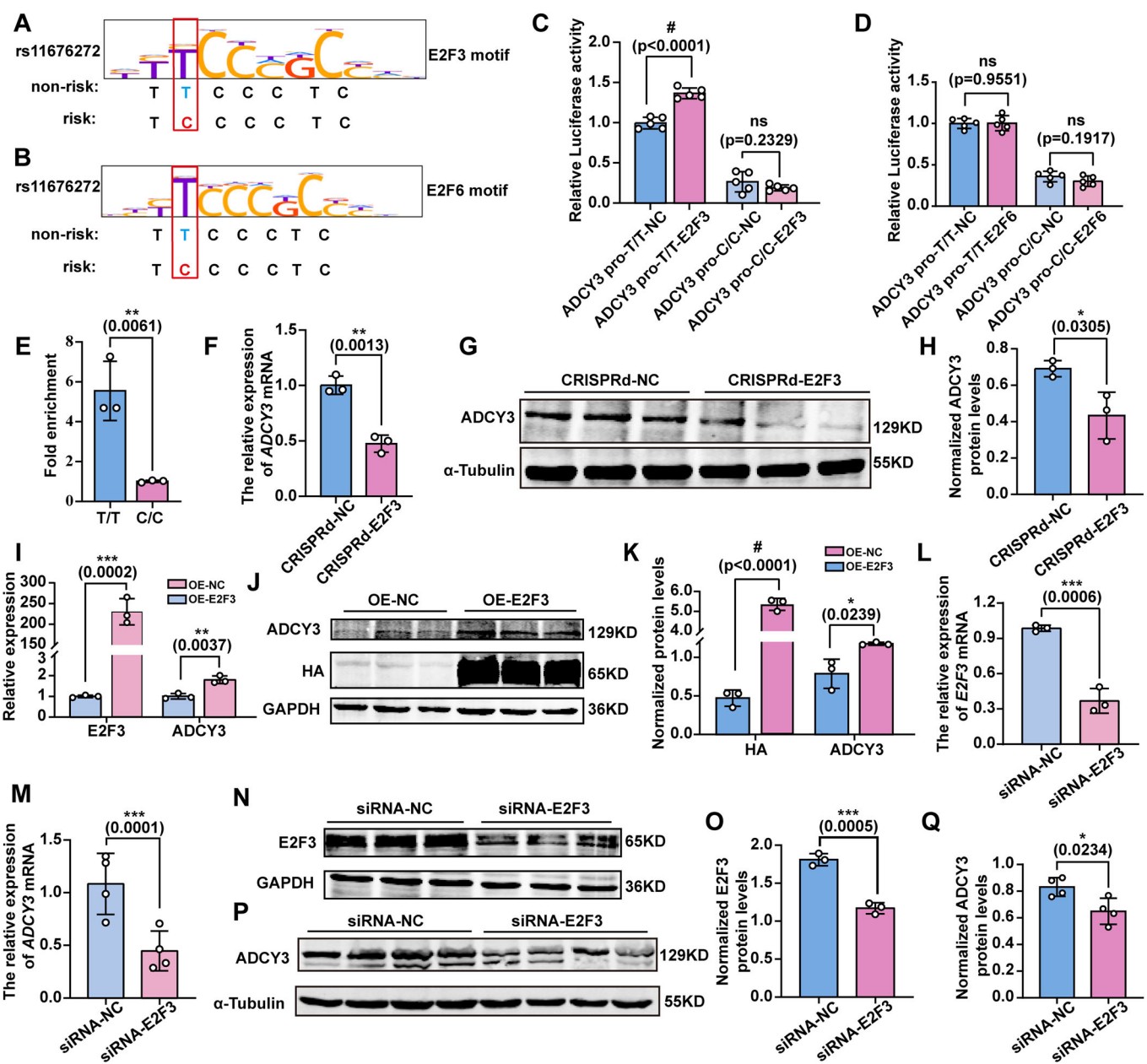

**Figure 3.  The rs11676272-C risk allele showed a significantly lower binding affinity of the transcriptional factor E2F3 to reduce the expression of ADCY3.**

(A, B) Schematics of E2F3 and E2F6 binding to the rs11676272-T nonrisk allele. (C, D) Dual-luciferase reporter assay detecting the binding affinity of E2F3 and E2F6 to the rs11676272-T and C alleles. The horizontal axis represents the grouping of alleles and TFs, and the vertical axis represents phase fluorescence activity values. $n = 5$ groups from three independent experiments. (E) ChIP-qPCR for *E2F3* at the DNA fragment containing the rs11676272-T nonrisk or -C risk alleles. $n = 3$ groups from three independent experiments. (F) qPCR detecting the *ADCY3* gene expression level in HEK293T cells transfected with CRISPRd-NC or CRISPRd-E2F3. $n = 3$ groups from three independent experiments. (G, H) Western blot analyzing ADCY3 level in HEK293T cells transfected with CRISPRd-NC or CRISPRd-E2F3. CRISPRd-NC: control group, CRISPRd-E2F3: inactivating the E2F3 binding site of the rs11676272 region. α-Tubulin was used as a loading control. $n = 3$ groups from three independent experiments. (I) qPCR determining E2F3 and ADCY3 expression levels in E2F3-overexpressing HEK293T cells. $n = 3$ groups from three independent experiments. (J, K) Western blot determining E2F3 and ADCY3 expression levels in E2F3-overexpressing HEK293T cells. OE-NC: control group, OE-E2F3: E2F3 overexpression group. GAPDH was used as a loading control. $n = 3$ groups from three independent experiments. (L) qPCR detecting the *E2F3* gene expression level in E2F3 knockdown cells. $n = 3$ groups from three independent experiments. (M) qPCR measuring *ADCY3* gene expression level in E2F3 knockdown cells. $n = 4$ groups from three independent experiments. (N, O) Western blot probing with anti-E2F3 in control and E2F3 knockdown cells. Control group: siRNA-NC, E2F3 knockdown group: siRNA-E2F3. GAPDH was used as a loading control. $n = 3$ groups from three independent experiments. (P, Q) Western blot probing with anti-ADCY3 in siRNA-NC and siRNA-E2F3 cells. α-Tubulin was used as a loading control. $n = 4$ groups from three independent experiments. All data were presented as mean ± SD. *$p < 0.05$, **$p < 0.01$, ***$p < 0.001$, #$p < 0.0001$ ("#" representing "*****"). ns no significance. *p* values were calculated using one-way ANOVA in (C, D), and Student's *t*-test in (E–Q). Source data are available online for this figure.

genome, to examine the allele-specific effects of rs11676272 on obesity. From this genetic background, we generated a mutant mouse line (Mut-hADCY3) with a targeted rs11676272 (T > C) variant via CRISPR-Cas9 and homologous recombination (homology-directed repair [HDR]) strategies (Fig. 4D). Sanger sequencing confirmed that the rs11676272-T protective allele had been edited to the C risk allele (Fig. 4E). Furthermore, Sanger sequencing analysis of the predicted possible off-target sites revealed no off-target editing in the Mut-hADCY3 mice.

## The rs11676272-C risk variant in hADCY3 knock-in mice leads to increased obesity under HFD feeding

To analyze the role of the rs11676272 locus in obesity in mice, we fed hADCY3 and Mut-hADCY3 male mice aged 4–5 weeks a standard chow diet (SCD) or an HFD. The body weights and food intake of the hADCY3 and Mut-hADCY3 mice were measured weekly until 18 weeks (Fig. 5A). Under SCD feeding, no significant differences in body weight or food intake between the two mouse lines were observed. However, under HFD feeding, the Mut-hADCY3 mice exhibited accelerated obesity and a significant increase in food intake in comparison with the hADCY3 mice (Fig. 5B–D).

Reportedly, GWAS-identified deleterious SNP variants within the hADCY3 gene are usually associated with a high BMI, total adiposity, and trunk fat mass (Speliotes et al, 2010). Accordingly, we investigated the impact of the rs11676272 locus on adiposity in male mice fed an HFD for 18 weeks. Compared with that in hADCY3 mice, weight gain in Mut-hADCY3 mice was accompanied by increased accumulation of white adipose tissue in both the subcutaneous and visceral areas (Fig. 5E). Metabolic cage experiments revealed that Mut-hADCY3 mice exhibited a reduced respiratory exchange ratio (RER) compared to hADCY3 mice, with no significant alterations in oxygen consumption ($VO_2$) or carbon dioxide production ($VCO_2$) (Fig. 5F–H). The total locomotor activity and neuronal activation in VMH and ARC were similar between hADCY3 and Mut-hADCY3 mice (Fig. EV6A–C). Moreover, compared with hADCY3 mice, Mut-hADCY3 mice presented significantly larger white adipocytes, smaller brown adipocytes at the scapulae, and larger lipid droplets in the liver (Fig. 5I,J). Collectively, the data indicated that the Mut-hADCY3 mice developed an accelerated obese phenotype under HFD feeding, exhibiting a significant increase in white adipose tissue and a significant decrease in brown adipose tissue compared with those of the hADCY3 mice.

## Mice carrying the hADCY3 rs11676272-C risk variant exhibit shorter cilia in vivo

The hypothalamus serves as the hub for metabolic homeostasis, playing an essential role in the regulation of body weight (Lei et al, 2024). We examined ADCY3 expression in the hypothalamic of Mut-hADCY3 and hADCY3 mice. qPCR and Western blot analysis revealed that ADCY3 expression in the hypothalamus was lower in Mut-hADCY3 mice than in hADCY3 mice (Fig. 6A–C). Reportedly, ADCY3 is localized mainly in neuronal cilia in the hypothalamus, and mice lacking ADCY3 exhibit shorter cilia in the VMH, which are responsible for regulating food intake and body weight (Yang et al, 2022). The VMH was selected for ciliary

analysis because it contains neurons with the longest cilia(Sipos et al, 2018; Sun et al, 2021), thereby enabling reliable cilium length measurement and effect assessment. Therefore, we quantified the percentage of ciliated cells and the length of the cilium in the VMH. The cilium length in the VMH of Mut-hADCY3 mice was significantly shorter than that in the VMH of hADCY3 mice, whereas there were no differences in the percentage of ciliated cells (Fig. 6D–F). Furthermore, we isolated mouse embryonic fibroblasts (MEFs) from hADCY3 and Mut-hADCY3 mice at 12.5–14.5 days of gestation. Western blot analysis revealed significantly lower ADCY3 expression levels in MEFs derived from Mut-hADCY3 mice than in those derived from hADCY3 mice (Fig. EV7A,B). Next, we quantified the percentage of ciliated cells and the cilium length in MEFs derived from hADCY3 and Mut-hADCY3 mice by staining for the cilium marker ARL13B. The results revealed that cilium formation did not differ, whereas cilium length was significantly shorter in Mut-hADCY3-derived MEFs than in cells derived from hADCY3 mice (Fig. EV7C–E). Taken together, these results indicate that the rs11676272-C risk variant reduces ADCY3 expression and impairs cilium growth in vivo.

To better understand whether the rs11676272 region exhibits enhancer activity in vivo, we activated the rs11676272 region in the VMH via the stereotactic delivery of an adeno-associated virus of CRISPRa (CRISPRa-AAV) into the VMH of hADCY3 and Mut-hADCY3 mice at age 4-5 weeks (Fig. 6G). Five weeks after the injection, the percentage of ciliated cells in the VMH remained unchanged, as revealed by staining with the ADCY3 antibody (Fig. 6H,I). Cilia were significantly longer in hADCY3 mice injected with CRISPRa-AAV than in those injected with the control vector (Fig. 6J). However, neither the percentage of ciliated cells nor the length of the cilium in the VMH was significantly altered in Mut-hADCY3 mice following CRISPRa (Fig. 6H–J). These findings indicate that the activation of the rs11676272 region by the CRISPRa approach increases ADCY3 expression, providing further evidence that rs11676272 possesses enhancer activity in vivo. In conclusion, we demonstrate that the rs11676272-T nonrisk allele exhibits enhancer activity in hADCY3 mice; however, the rs11676272-C risk variant exhibits significantly reduced enhancer activity, resulting in decreased ADCY3 expression and shorter cilia.

## Activation of the ADCY3 promoter in the VMH and ARC partially rescues the shortened cilia in Mut-hADCY3 mice

CRISPR-mediated activation of promoters or enhancers has been reported to be effective in rescuing against haploinsufficiency-induced obesity by increasing gene expression levels (Matharu et al, 2019). However, we found that activation of the enhancer activity of the rs11676272 region failed to restore either the ADCY3 expression level or the length of the shortened cilia in Mut-hADCY3 mice (Fig. 6). Therefore, we sought to examine whether the shorter cilia and obese phenotype observed in Mut-hADCY3 mice could be alleviated by activating the promoter region of hADCY3. To do this, we designed two sgRNAs targeting the ADCY3 promoter and individually cloned them into the pEASY-U6 vector. Then, we transfected HEK293T cells with Pro-NC, Pro-sgRNA1, Pro-sgRNA2, and Pro-sgRNA1 + 2 along with the CRISPRa construct. qPCR and western blotting revealed that the combination of two sgRNAs had greater effects, increasing the

endogenous *ADCY3* mRNA and protein levels by fourfold (Fig. 7A) and twofold (Fig. 7B,C), respectively. Therefore, CRISPRa-sgRNA1 + 2 was used for subsequent validation.

The ARC and VMH play central roles in regulating energy homeostasis, with the *ADCY3* gene highly expressed in these two nuclei (Yan et al, 2025; Yang et al, 2022). To assess the efficacy of CRISPRa in ameliorating obesity in Mut-hADCY3 mice, we stereotaxically administered CRISPRa, which targets the *hADCY3* promoter, into the VMH and ARC of 4–5-week-old Mut-hADCY3 mice (Fig. 7D). The body weights and food intake of the control and CRISPRa-injected (Pro-CRISPRa) groups did not significantly differ over the 8 weeks of treatment (Fig. 7E,F). Furthermore, we quantified the percentage of ciliated cells and the cilium length in the VMH and ARC by staining for the neuronal cilium marker ADCY3. The percentages of ciliated cells did not differ between the control and CRISPRa-injected mice, whereas the cilium length was significantly longer in the Mut-hADCY3 mice injected with the Pro-CRISPRa virus than in the control group mice (Fig. 7G–I). Moreover, to exclude the activation of the mouse *Adcy3* promoter by Pro-CRISPRa as a potential explanation for these effects, we delivered the Pro-CRISPRa virus into the VMH and ARC of 4–5-week-old wild-type mice, in which the *hADCY3* promoter does not exist. We found that neither the percentage of ciliated cells nor the cilium length significantly changed (Fig. EV8A–C), thus ruling out the role of mouse-derived Adcy3. Collectively, our results demonstrate that upregulation of hADCY3 expression can rescue the shortened cilia phenotype in Mut-hADCY3 mice. Given that the Mut-hADCY3 mouse model involves a whole-body point mutation but that the upregulation of hADCY3 expression occurs exclusively in the VMH and ARC, it is reasonable that the obese phenotype of the mice was not rescued.

## Discussion

In this study, we demonstrated that rs11676272 is located within an enhancer region of the *ADCY3* gene, where the C risk allele disrupts E2F3 binding, leading to reduced *ADCY3* promoter activity and lower ADCY3 expression levels. Furthermore, we generated Mut-hADCY3 mutant mice via the CRISPR–Cas9 strategy, which exhibited significantly reduced enhancer activity in the rs11676272 region and developed an obese phenotype under a HFD. Additionally, activating the *hADCY3* promoter region restored ADCY3 expression and ameliorated the shortened ciliary phenotype in Mut-hADCY3 mice. Taken together, our data highlight the allele-specific role of rs11676272 in modulating *ADCY3* expression and obesity pathogenesis.

Enhancers regulate gene expression in metazoans, orchestrating the spatiotemporal activation of transcriptional genes by interacting with promoters (Yang and Hansen, 2024). Typically, when promoters and enhancers are in linear proximity (spanning thousands to tens of thousands of base pairs), they can establish interactions along the chromatin without requiring three-dimensional proximity (Furlong and Levine, 2018). These interactions may involve transcription machinery, such as RNA polymerase II recruitment, or protein-mediated interactions via transcription factor binding (Bulger and Groudine, 1999; Moreau et al, 1981). Our findings demonstrate that the regulatory effect of rs11676272 on *ADCY3* requires the participation of the E2F3 TF,

suggesting that the rs11676272 region might facilitate functional connections with the *ADCY3* promoter region through E2F3 binding. Nevertheless, many enhancers can bypass the nearest promoter to activate more distal promoters (Chen et al, 2024; Li et al, 2012). Therefore, we cannot exclude the possibility that the rs11676272 region also regulates the activity of the promoters of distal genes. This hypothesis merits further investigation via chromosome conformation capture (3 C) sequencing assays or related genomic assays in subsequent studies.

Reportedly, a variation in the promoter of the apolipoprotein A-II gene (*APOA2*), -265 T > C (rs5082), results in different responses to saturated fatty acids depending on the allele at this site. Among individuals with low saturated fatty acid levels, carriers of neither the T allele nor the C allele of rs5082 developed obesity. However, when highly saturated fatty acids are consumed, individuals with the C allele exhibit a significantly increased risk of obesity compared with those with the T allele (Corella et al, 2007; Corella et al, 2011; Gkouskou et al, 2024). These findings are consistent with our observation that the body weights of hADCY3 and Mut-hADCY3 mice differed significantly only under HFD conditions. These findings suggest that the rs11676272 locus is particularly susceptible to a HFD. Interestingly, a diet with moderately high protein intake (30% of energy from proteins) can attenuate the effects of *FTO* variants on obesity (Gkouskou et al, 2024; Merritt et al, 2018). Thus, appropriately adjusting dietary structure can help avoid a range of metabolic diseases, including obesity, caused by genetic variations. Another study involving 147 overweight or obese individuals revealed that *ADCY3* gene variants differentially respond to nutrients. Individuals carrying the rs10182181-G allele were more susceptible to a low-protein diet than to a moderately high-protein diet (Goni et al, 2018). Therefore, exploring the role of genetic variants in the *ADCY3* gene in an individual context may facilitate the development of personalized nutritional management strategies for obesity and other diet-related pathologies.

The severity of obesity is reportedly regulated by the dosage of specific genes, such as *FTO* and *MC4R*. Compared with mice carrying two copies of the *FTO* gene, mice carrying 3-4 copies are more obese (Church et al, 2010). Mice with *MC4R* haploinsufficiency develop obesity due to polyphagia, hyperinsulinemia, and hyperglycemia (Matharu et al, 2019). Like those of *FTO* and *MC4R*, the *ADCY3* expression level also has a reverse dose-dependent effect on obesity development; mice with *ADCY3* gene knockout develop severe obesity (Wang et al, 2009). Additionally, haploinsufficiency renders mice susceptible to HFD-induced obesity (Tong et al, 2016). In contrast, mice with an additional copy of hADCY3 resist HFD-induced obesity (Yang et al, 2022). In gene therapy, obesity can be treated by upregulating endogenous gene expression (Matharu et al, 2019). Our results demonstrate that targeting the *ADCY3* promoter region could partially rescue the ciliary shortening phenotype, suggesting that SNP variants causing gene expression dosage alterations may be therapeutically addressed by targeting their promoter regions to alleviate disease phenotypes.

Although we hypothesize that ciliary shortening is an important factor contributing to obesity, we cannot exclude additional mechanisms. Previous studies have shown that ADCY3 plays a crucial role in BAT, and the loss of its variant ADCY3-AT enhances energy expenditure and protects against obesity and metabolic disorders (Khani et al, 2024). Moreover, ADCY3 is the

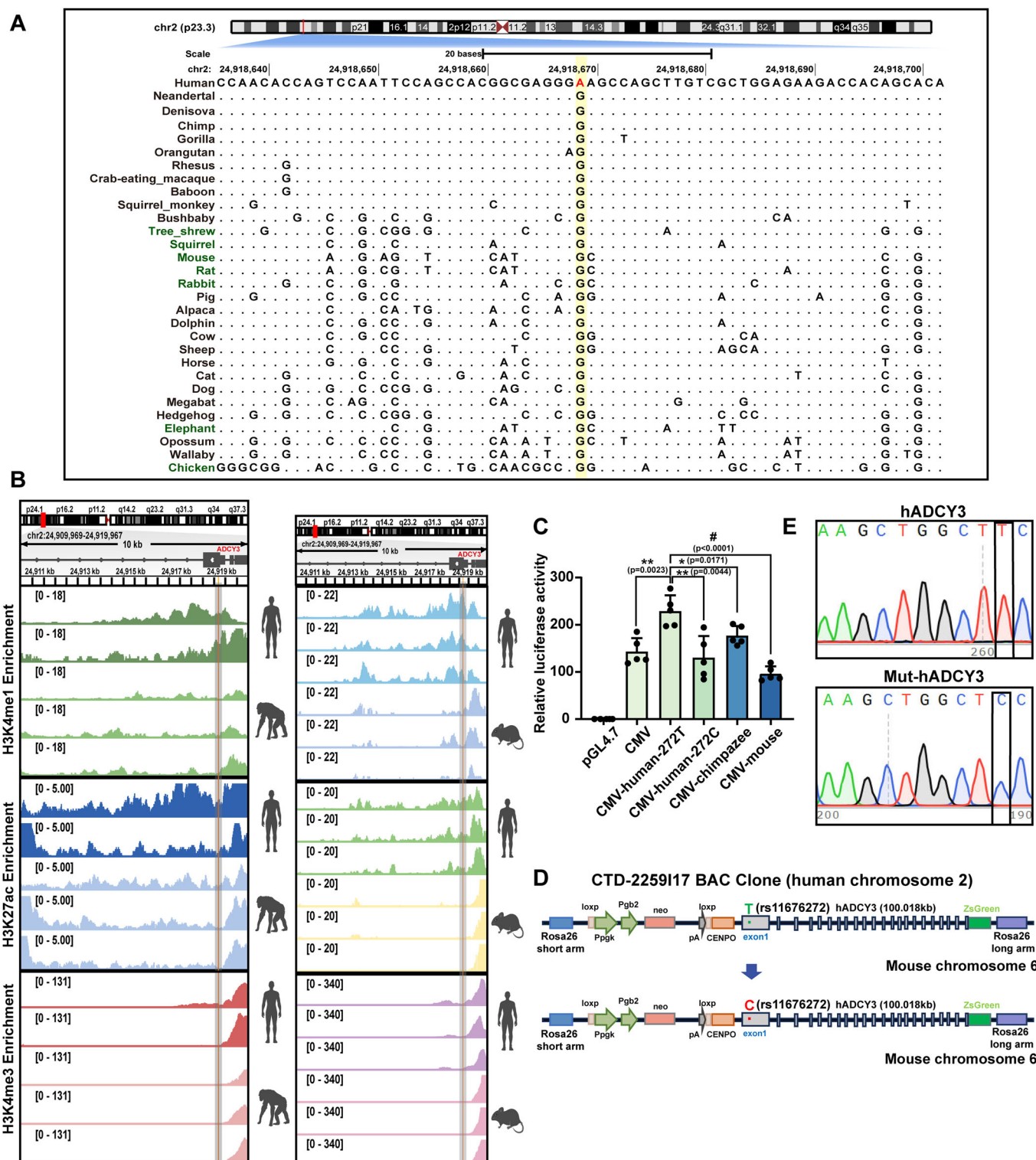

predominant adenylate cyclase isoform expressed in WAT, and its mutation markedly reduces adenylate cyclase activity in this tissue (Pitman et al, 2014). Consistent with these findings, Mut-hADCY3 mice developed pronounced hepatic steatosis. It has been reported that ADCY3 activity is reduced in the liver of ob/ob mice, and

liraglutide treatment significantly upregulates hepatic ADCY3 expression in obese mice, suggesting a potential role of hepatic ADCY3 in modulating the obesity phenotype (Liang et al, 2016); In addition, GPR75-deficient mice are resistant to HFD-induced obesity primarily attributed to alterations in feeding behavior;

◄ **Figure 4. The rs11676272-C risk variant mouse model in human *ADCY3* knock-in mouse.**

(A) UCSC multiz alignment. The red letter highlights the rs11676272 locus in human, and the position derived from human is outlined in yellow. Dots indicate bases identical to human. (B) ChIP-seq tracks of RPKM normalized H3K4me1, H3K27ac and H3K4me3 enrichment in different tissues from human, chimpanzee and mouse samples across 10 kb region within the *ADCY3* gene. The rs11676272 loci is highlighted in orange. Prediction of enhancer activity of rs11676272 homologous sequence of the *ADCY3* gene in the mouse and chimpanzee using ENCODE and UCSC databases. (C) The dual-fluorescence reporter system detected the enhancer activity of the rs11676272 homologous sequence on the mouse and chimpanzee *ADCY3* gene. $n = 5$ groups from three independent experiments. (D) Human *ADCY3* gene insertion in hADCY3 mice. The green "T" represents the rs11676272 SNP locus; Mutation of rs11676272 SNP locus in Mut-hADCY3 mice. The red "C" indicates a mutated base at the rs11676272 SNP locus. (E) Sanger sequencing of hADCY3 and Mut-hADCY3 mouse. The box indicates the rs11676272 locus. All data were presented as mean ± SD; *$p < 0.05$, **$p < 0.01$, #$p < 0.0001$ and $p$ values were calculated using one-way ANOVA. Source data are available online for this figure.

however, the authors also noted that cilia-related signaling may contribute to dietary responses (Jiang et al, 2024). Similarly, we observed increased food intake in Mut-hADCY3 mice. Thus, while the rs11676272 variant promotes obesity in part through ciliary dysfunction, the altered feeding behavior and impaired ADCY3 signaling in adipose tissue cannot be ruled out.

Our data demonstrate that coding variants influence the risk of obesity by modulating TF binding and gene expression. We revealed that E2F3, a multifaceted TF, preferentially binds to the rs11676272-T allele and increases the ADCY3 expression level. E2F3 belongs to the E2F family and is highly expressed in cells (Inoshita et al, 1999; Kassab et al, 2023). Mice lacking E2f3 in muscle are susceptible to obesity and diabetes under HFD conditions, which implicates E2f3 in metabolism (Bahn et al, 2023). In addition, E2F3 is involved in regulating the length of the G0/G1 phase of the cell cycle (Zhu et al, 2004), which is the phase during which cilium formation occurs. Although the relationship between E2F3 and cilia has not been reported, several members of the E2F family, including E2F4, E2F5, and E2F7, have been implicated in cilium formation (Choksi et al, 2024; Hazan et al, 2021; Xie et al, 2020). Thus, E2F3 may form a complex with other family members to affect cilium length and regulate the occurrence of obesity.

The interpretation of GWAS signals is frequently complicated by extensive LD, which can create dense clusters of associated variants without reflecting true causal complexity (Schaid et al, 2018). At the *ADCY3* locus, LD-aware fine-mapping resolves this apparent heterogeneity, supporting a parsimonious genetic model in which a single variant drives the obesity association. The exclusive inclusion of rs11676272 in the 95% credible set (PIP >0.999), together with the monotonic decay of association strength with decreasing LD, provides compelling evidence for a single major signal and effectively excludes independent contributions from other linked common variants. Crucially, this inference is reinforced by in vivo evidence showing that rs11676272 alone is sufficient to induce an obesity phenotype (Fig. 5). More broadly, these findings underscore the importance of integrating LD deconvolution with functional validation to translate GWAS associations into causal mechanisms with clear biologically interpretability.

Notably, our study focused on the regulatory role of the rs11676272 mutation in obesity phenotypes in male mice, whereas the metabolic effects in female individuals remain uninvestigated. Previous studies have shown that *Adcy3* gene deficiency induces obesity phenotypes in both female and male mice (Wang et al, 2009). Compared with wild-type controls, *Adcy3* heterozygous null mice of both sexes exhibited obesity under HFD feeding

(Tong et al, 2016). Therefore, we infer that *ADCY3* does not exhibit sex differences in obesity regulation. The use of male mice in this study was primarily based on the following considerations: first, hormonal fluctuations during the female reproductive cycle may affect metabolic parameters, increasing experimental variability; second, female mice are more resistant to HFD-induced obesity (Oraha et al, 2022), potentially obscuring the effects of genetic variations on energy balance.

It should be noted that this study has several limitations. First, rs11676272 is a missense variant located within the coding region of ADCY3 and could, in principle, alter protein structure and thus affect enzymatic activity. Although AlphaFold structural predictions and enzymatic assays detected no significant alterations, our inability to purify the ADCY3 protein prevents us from entirely excluding an effect on enzymatic activity. Nevertheless, consistent evidence from cellular and in vivo experiments clearly establishes that rs11676272 resides in an enhancer region and modulates ADCY3 expression by modulating enhancer activity. Accordingly, the central conclusion of our study—that rs11676272 regulates ADCY3 expression through an enhancer-mediated mechanism—remains firmly supported, irrespective of its potential effects on enzymatic activity. Second, although multiple hypothalamic nuclei are involved in metabolic regulation, pinpointing the specific nuclei responsible for the observed phenotype remains challenging. As a result, our rescue experiments targeted the VMH and ARC primarily for technical convenience and the reliability of ciliary length measurements in these regions. Third, the downstream pathways by which the rs11676272 variant promotes obesity are not yet fully elucidated and will be the focus of future research.

In conclusion, by combining bioinformatics analyses, experimental validation, and in vivo functional assays, we demonstrated that rs11676272 functions as an allele-specific enhancer to regulate *ADCY3* expression and obesity through E2F3 binding. These findings highlight rs11676272 as a promising therapeutic target for treating obesity.

## Methods

**Reagents and tools table**

| Reagent/ resource | Reference or source | Identifier or catalog number |
| --- | --- | --- |
| **Experimental models** | | |
| HEK293T | Procell | CL-0005 |
| SY5Y | Procell | CL-0208 |

| Reagent/ resource | Reference or source | Identifier or catalog number |
|---|---|---|
| Lo2 | Procell | HL-7702 |
| 3T3-L1 | Procell | CL-0006 |
| hADCY3 | Shanghai Model Organisms | BL6N/129-rs11676272-T |
| Mut-hADCY3 | GemPharmatech | BL6N/129-rs11676272-C |
| **Recombinant DNA** | | |
| pSpCas9(BB)-2A-Puro (PX459) | Addgene | 48139 |
| pGL4.54 | Promega | E5061 |
| pGL4.70 | Promega | E6881 |
| CRISPRa | VectorBuilder | VB211219 |
| CRISPRi | Addgene | 71236 |
| pAM_dCas9 | Active Motif | 53122 |
| pLV3-CMV-MCS-FLAG-Puro | NovoPro | V014731 |
| **Antibodies** | | |
| E2F3 | Proteintech | 27615-1-AP |
| ADCY3 | Bioss/Invitrogen/Novus | bs-2027R/PA5-35382/ NBP1-92683 |
| ARL13B | Proteintech | 17711-1-AP |
| α-Tubulin | Sigma | AC035 |
| β-actin | TransGen Biotech | HC201-01 |
| HA tag | Abcam | ab236632 |
| GAPDH | TransGen Biotech | HC301-01 |
| Secondary antibodies-Rabbit | SeraCare KPL | 5230-0416 |
| Secondary antibodies-Mouse | SeraCare KPL | 5230-0412 |
| **Oligonucleotides and other sequence-based reagents** | | |
| PCR primers | This study | Table EV2 |
| **Chemicals, enzymes and other reagents** | | |
| T7 Endonuclease I | New England BioLabs | M0318L |
| FastDigest BbsI (BpiI) | Thermo Scientific | FD1014 |
| T4 polynucleotide kinase | New England BioLabs | M0201S |
| T4 DNA Ligase | Takara | 2011A |
| BamHI | New England BioLabs | R0136V |
| KpnI | New England BioLabs | R3142V |
| **Software** | | |
| IGV | IGV: Integrative Genomics Viewer | |
| GraphPad Prism | Prism - GraphPad | |
| EndNote | EndNote - The Best Citation & Reference Management Tool | |
| AlphaFold Server | AlphaFold Server | |
| **Other** | | |

## Annotation of candidate SNPs

The dataset of hg38: 24,819,168–24,920,237 was obtained from the meta-analysis of BMI in UK Biobank and GIANT database (see Dataset EV1 for details). An in-depth screen was conducted to identify the obesity-associated SNP loci ($p < 1 \times 10^{-8}$). Then, the functional SNPs were annotated from functional genomes by RegulomeDB (Boyle et al, 2012), which involves multiple evidence-based approaches including ENCODE transcription factors, histone CHIP-seq, FAIRE, and eQTL. Lower rankings indicate higher functional probability. Finally, an R package comprising dplyr, BioSeqUtils, and ggplot2 were used to visualize the data. The source data are listed in Dataset EV1.

## LD analysis and statistical fine-mapping

To evaluate the 123 SNPs within the *ADCY3* gene region, pairwise LD statistics were computed for all variants using LDlink (Machiela and Chanock, 2015), with the 1000 Genomes Project Phase 3 European population as the reference panel. Statistical fine-mapping was conducted using the approximate Bayes factor (ABF) method to calculate the posterior inclusion probability (PIP) for each variant. The prior variance was set to 0.04, consistent with typical effect sizes for quantitative traits. The 95% credible set was defined as the smallest set of variants whose cumulative PIP exceeded 0.95. The source data are listed in Dataset EV2.

## AlphaFold-based structural modeling of hADCY3

Three-dimensional structures of ADCY3 carrying the rs11676272-T and -C allele were predicted using the AlphaFold Server. The structural alignment was performed in the PyMOL using conserved residues as a reference to ensure alignment accuracy. The overall root-mean-square deviation (RMSD) between the aligned structures was calculated to assess the structural perturbation of this SNP.

## ChIP-seq datasets analysis

The ChIP-seq datasets used in this manuscript are accessible from the European Nucleotide Archive and ENCODE repositories (Luo et al, 2020). The details including specific accession numbers are provided in Table EV1. The Raw fastq files were trimmed using Trim Galore v0.6.10 and mapped to the human (hg38) and chimpanzee (panTro5) genomes using Bowtie2 v2.5.4. ChIP-seq peaks were identified by MACS3 (Zhang et al, 2008) on BAM files with default parameters, with more stringent settings applied for specific histone marks, including "--broad-cutoff 0.05" for broad marks (H3K4me1) and "-q 0.01" for narrow marks (H3K4me3 and H3K27ac). To remove background noise, the bdgcmp command was employed to compare the bedgraph files of the control and treatment via generating bigwig files directly. To directly visualize the differences in signal enrichment at homologous SNP sites across species, the panTro5ToHg38.over.chain.gz and mm10ToHg38.over.chain.gz files were downloaded from the UCSC genome browser and mapped to the human genome (hg38) using CrossMap v0.7.3 (Zhao et al, 2014).

## Cell culture and transfection

The HEK293T, 3T3-L1 and SY5Y cell lines were obtained from Pricella and cultured in DMEM (Gibco) supplemented with 10% fetal bovine serum (FBS) and 1% penicillin/streptomycin (Gibco). The LO2 cell lines was obtained from Pricella and cultured in RPMI-1640 (Gibco) supplemented with 10% FBS and 1% penicillin/streptomycin. The mouse embryonic fibroblasts (MEFs) were derived from the hADCY3 and Mut-hADCY3 mice. Briefly, pregnant mice on 12.5–14.5 days were euthanized by cervical dislocation and disinfected with 75% alcohol. The stomachs were incised, and the fetal mice were carefully removed. Then, the limbs, head, and tail were excised, leaving only the trunk. Following three washes with PBS, 0.25% trypsin was added, and the cells were digested at 37 °C. Thereafter, cell culture was carried out in a T25 bottle.

## Animals

All experimental procedures used in the study were conducted in accordance with the Guiding Opinions on the Treatment of Experimental Animals. The hADCY3 knock-in mice were previously generated (Yang et al, 2022). The Mut-hADCY3 mice were constructed from the original hADCY3 mice. In brief, 20 female hADCY3 mice aged 3–4 weeks were bred with three male hADCY3 mice aged 12–15 weeks. Then, fertilized eggs were collected and injected with the Cas9, sgRNA, and a homologous recombination vector. The surviving fertilized eggs were then transferred into the uterus of a surrogate female mouse. The F0 mice were subjected to genomic DNA extraction for genotyping and screening for the mice with the correct mutation at the rs11676272 locus.

## Viruses and stereotactic injection of AAV

The following viral constructs were used: AAV2/9-miniCMV-sp_dCas9-VP64 and AAV2/9-U6-sgRNA-CMV-EGFP-P2A-MS2-P65-HSF1 (titer: 1E + 13 vector genome (v.g.)/mL respectively, PackGene Biotech). The hADCY3 and Mut-hADCY3 mice aged 5 weeks were lightly anesthetized with isoflurane (1–2%). For rs11676272 region activation, the AAV2/9-miniCMV-sp_dCas9-VP64 and AAV2/9-U6-sgRNA-5-CMV-EGFP-P2A-MS2-P65-HSF1 were mixed in a 1:1 ratio, and a total of 100 nL mixture was injected into the VMH (ML: ±0.4 mm, AP: −1.45 mm, DV: −5.35 mm) using a stereotaxic approach. For ADCY3 promoter activation, the AAV2/9-miniCMV-sp_dCas9-VP64 and AAV2/9-U6-sgRNA1 + 2-CMV-EGFP-P2A-MS2-P65-HSF1 were mixed in a 1:1 ratio, and a total of 120 nL mixture was injected into the VMH and ARC (ML: ±0.3 mm, AP: −1.45 mm, DV: −5.45 mm) using a stereotaxic approach. Immunofluorescence analysis was performed five weeks after injection.

## Dual-luciferase reporter assay

The luciferase construct was generated by inserting a 518-bp DNA fragment containing rs11676272 along with the ADCY3 promoter (~2000 bp) into the pGL4.7 vector (Promega, Cat: E6881) using the pEASY-Basic Seamless Cloning and Assembly Kit (TRAN, Cat: CU201-02). The 518-bp fragment containing rs11676272 was used as the control. The rs11676272-C risk allele was introduced into the luciferase constructs via the Mut Express II Fast Mutagenesis Kit V2 (Vazyme, Cat: C214-01). All plasmids were confirmed by Sanger sequencing.

HEK293T cells ($2 \times 10^4$/well) were plated in 96-well plates 12 h before transfection. The reporter vectors were cotransfected with the internal reference pGL4.54 vectors using Lipofectamine™ 3000 Reagent (L3000015, Invitrogen) according to the manufacturer's instructions. Following 48 h, the cells were harvested, and the luciferase activity was determined using the Dual-Luciferase Reporter Assay System according to the manufacturer's instructions (Promega, Cat: E2920).

## Genome editing in cell lines

To knock out the rs11676272 locus, we applied the CRISPR-associated RNA-guided Cas9 (CRISPR-Cas9) strategy. In brief, sgRNAs were designed from the first exon of the human ADCY3 gene with CRISPROR (http://crispor.tefor.net/) according to the protocols (Ran et al, 2013). Five pairs of sgRNA (each pair including sgRNA-top and sgRNA-bottom) were selected and synthesized. Next, 1 μL sgRNA-top (100 mM) and 1 μL sgRNA-bottom (100 mM) were annealed in 1 × T4 ligation buffer with 1 μL T4 PNK. Subsequently, the oligonucleotides were inserted into the pSpCas9 (BB)-2A-Puro (PX549) V2.0 plasmid (Addgene, 48139). Following 48-hour selection by puromycin, genomic DNA was extracted for PCR amplification of target fragments. The PCR products were digested using T7E1, and insertion/deletion (InDel) rates were calculated. The optimal pair of sgRNA, which was reflected by the highest InDel rate, was selected for generating cell clones with the knockout of the rs11676272 SNP locus.

To generate the rs11676272-C risk variant cell lines, the cells were cotransfected with the optimal sgRNA and an oligonucleotide (~100 bp) containing T > C mutation (1:1) into HEK293T cells. The medium was changed 24 h after transfection, and 10 μg/mL puromycin (Sigma) was added 48 h after transfection. Following the removal of the non-transfected cells, the surviving cells were trypsinized and sorted using FACS to establish single-cell clones. The single cells were seeded into 96-well plates and monitored over 9–14 days to exclude the non-single clones. Finally, the single clones were selected for subculture and genotyping.

## Off-target sequence analysis

The off-target sites for the rs11676272 region were predicted using CRISPROR (Haeussler et al, 2016). The top ten sites were ranked by the mitOfftarget Score and were sequenced by to Sanger sequencing. In brief, the whole genomic DNA of HEK293T rs11676272-C cells and Mut-hADCY3 mouse tails were extracted, and the predicted off-targets were amplified using PCR. The PCR products were purified or gel-recovered followed by Sanger sequencing. T obtained sequences were aligned to the wild-type sequences by blast to detect the off-target effect of CRISPR-Cas9.

## Detection of ADCY3 enzymatic activity

ADCY3 enzymatic activity was measured according to the manufacturer's protocol (RENJEBIO, RJ14314). Standard curves were generated using recombinant ADCY3 at 0, 50, 100, 200, 400, and 800 pg/mL. Sample wells received 10 μL of cell lysates plus 40 μL of diluent, while blank wells contained

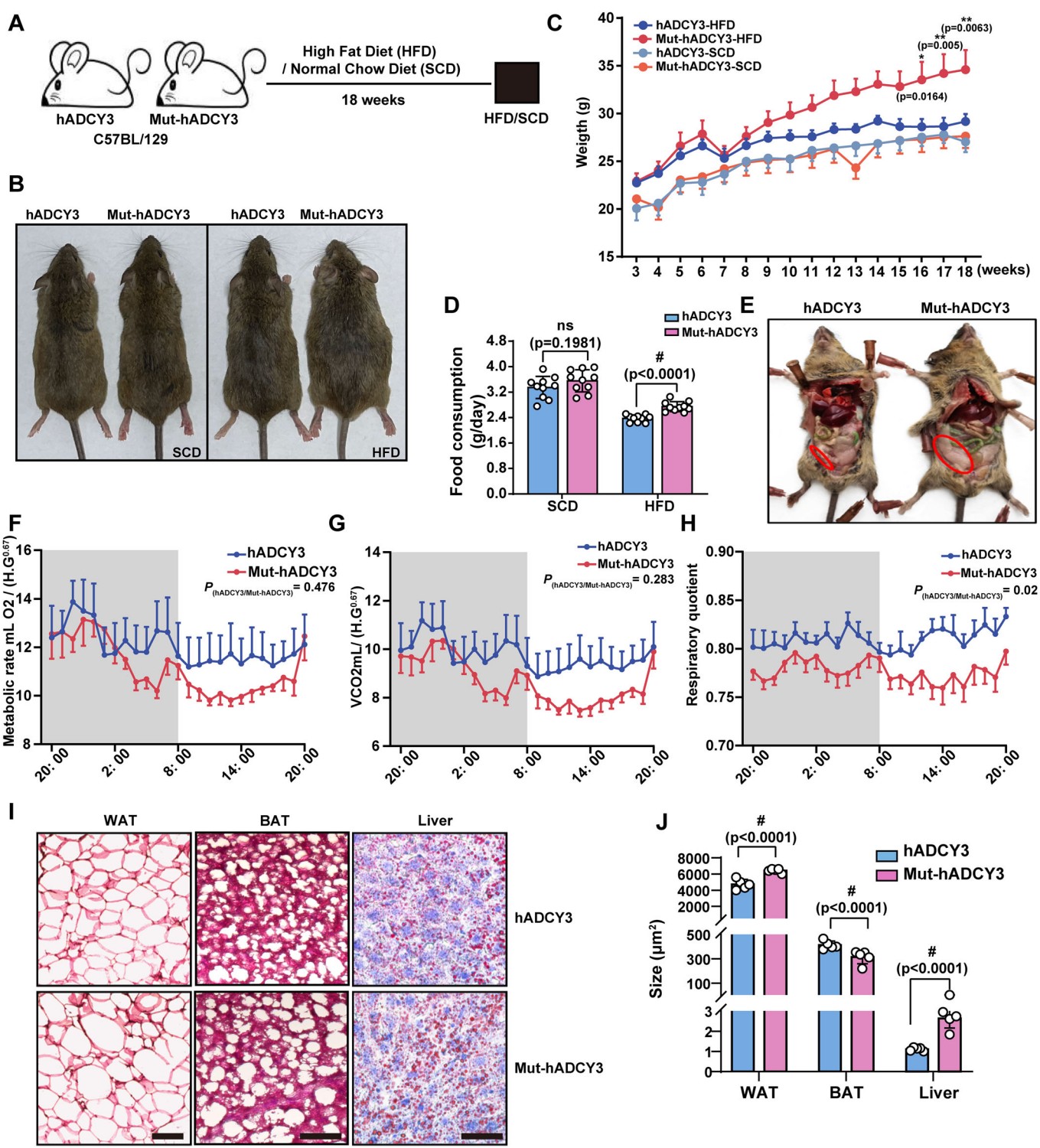

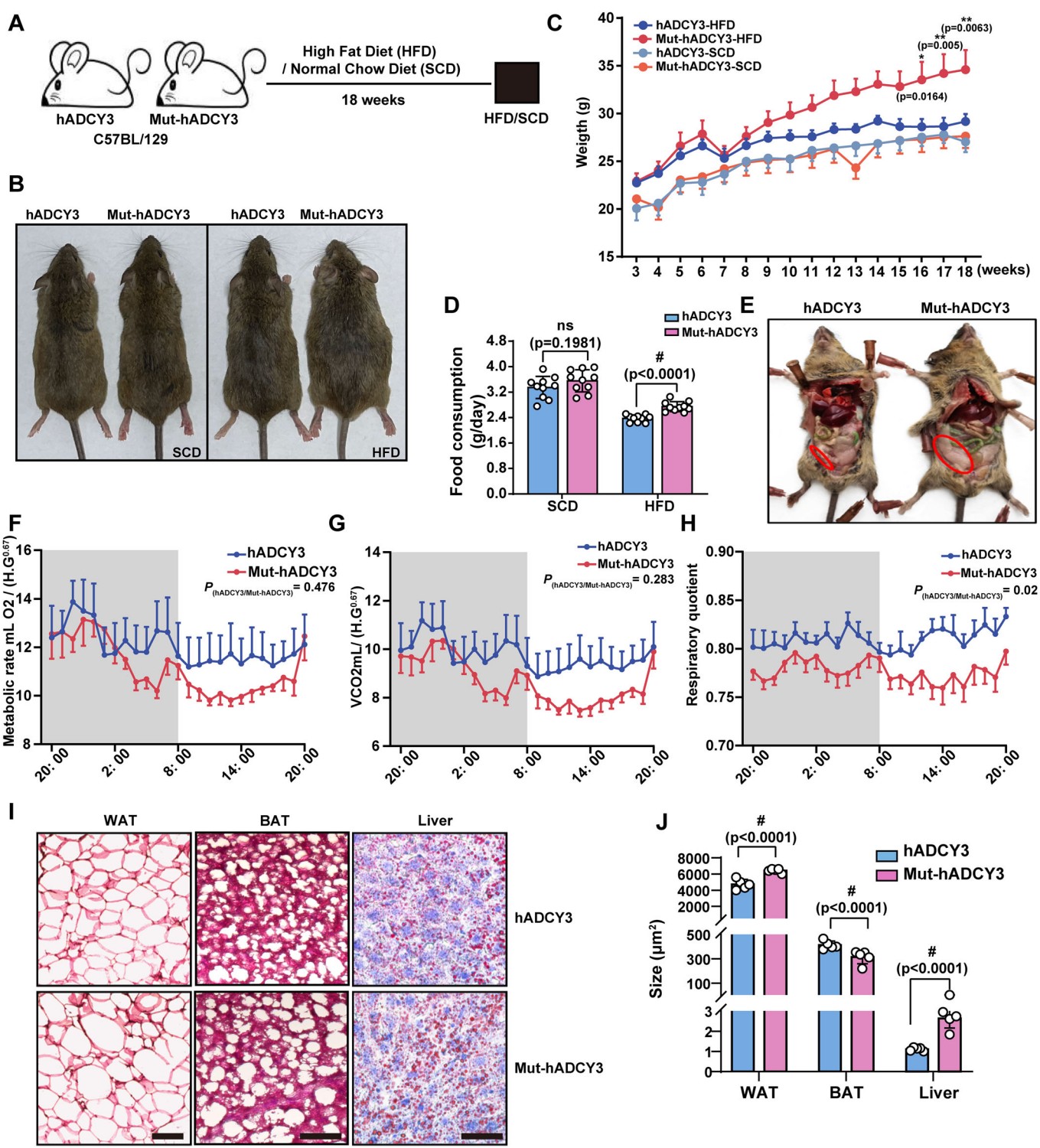

diluent only. Then, 100 µL HRP-conjugated detection antibody was added to each well except blanks. Plates were sealed and incubated at 37 °C for 60 min. Wells are then aspirated, patted dry, and washed five times with wash buffer (1 min per wash). Next, 50 µL each of Substrates A and B was added, and plates were incubated in the dark at 37 °C for 15 min. The reaction was terminated with 50 µL stop solution, and absorbance at 450 nm was measured within 15 min.

## cAMP assay

The levels of cAMP production were determined at 37 °C in HEK293T cells of rs11676272-T/T (T/T) and rs11676272-C/C (C/C). The cells were washed in Krebs–Ringer buffer (KRB) (135 mM NaCl, 3.6 mM KCl, 5 mM NaHCO$_3$, 0.5 mM NaH$_2$PO$_4$, 0.5 mM MgCl$_2$, 10 mM HEPES, 1.5 mM CaCl$_2$, 5 mM glucose, 0.1% BSA; pH 7.4) and subsequently incubated for 30 min in KRB in the

**Figure 5.  The rs11676272-C risk variant of hADCY3 knock-in mice exhibits accelerated obesity under HFD feeding.**

(A) Feeding diagram of the hADCY3 and Mut-hADCY3 mice to determine the effect of rs11676272-C risk variant of ADCY3 on body weight under SCD and HFD. (B) Representative images of the hADCY3 and Mut-hADCY3 mice following 18 weeks of SCD and HFD. (C) Quantification of body weights of the hADCY3 and Mut-hADCY3 mice under SCD or HFD. $n = 10$ groups from three independent experiments. (D) Quantification of food consumption of hADCY3 and Mut-hADCY3 mice under SCD or HFD. $n = 10$ groups from three independent experiments. (E) Representative images of subcutaneous fat accumulation in hADCY3 and Mut-hADCY3 mice. The red circles indicate fat accumulation. (F–H) $VO_2$ (F), $VCO_2$ (G), and RER (RER $= VCO_2/VO_2$) (H) in hADCY3 and Mut-hADCY3 mice. $n = 6$ groups from three independent experiments. (I) Representative images of hematoxylin and eosin (H&E) staining of the white adipose tissue (WAT), brown adipose tissue (BAT), and liver from the hADCY3 and Mut-hADCY3 mice under HFD. Scale bars, 50 μm (WAT and BAT), 100 μm (Liver). (J) The average size of adipocytes and liver oil droplets in the hADCY3 and Mut-hADCY3 mice under HFD. $n = 5$ groups from three independent experiments. All data were presented as mean ± SD. $*p < 0.05$, $**p < 0.01$, $\#p < 0.0001$ ns, no significance. $p$ values were calculated using repeated measures ANOVA, with Bonferroni correction in F to H and two-way ANOVA in (C, D, J). Source data are available online for this figure.

presence or absence of 2 μM forskolin and 0.1 mM IBMX (Roger et al, 2011). Next, the KRB was aspirated, and the cells were solubilized with 0.1 M HCl and 1% Triton X-100 (300 μL/well) at 37 °C for 30 min, followed by centrifugation at $600 \times g$ for 5 min at 4 °C. The supernatants were either stored at $-80$ °C or assayed directly using the Cyclic AMP Direct EIA Kit (Arbor Assays). The protein concentration in the supernatant was determined by bicinchoninic acid (BCA) assay (Vazyme, E112-02).

## CRISPRa, CRISPRi, and CRISPRd

The U6 enhancer and promoter, as well as sgRNA scaffold from the pX459 vector were cloned into the *pEASY-Blut-Zero* vector (*pEASY*, #CB501-01, TRAN) to prepare the *pEASY-U6-gRNA* expression construct. The sgRNAs targeting the rs11676272 region or the *ADCY3* promoter region were designed using http://crispor.tefor.net/, and subsequently cloned into the BbsI of the *pEASY-U6-gRNA* expression construct (TRAN, CB501). HEK293T cells were transfected with the *pEASY-U6-gRNA* expression construct and CRISPRa (VB211219-1067nmf, VectorBuilder), CRISPRi (Addgene, 71236), or CRISPRd (Active Motif, 53122) vectors using Lipofectamine™ 3000 Reagent (L3000015, Invitrogen) according to the manufacturer's instructions.

## Electrophoretic mobility shift assay (EMSA)

Nuclear extracts from HEK293T cells were prepared using the nuclear protein extraction kit (Beyotime, Cat. No. P0028). The nuclear extract proteins were incubated with the biotin-labeled DNA probe (Beyotime, Cat. No. GS008) containing rs11676272 region. The complexes were detected with the chemiluminescence EMSA kit (Beyotime, Cat. No. GS009), followed by electrophoresis and membrane transfer. A super-shift assay was performed by adding an antibody specific to particular TFs.

## Chromatin immunoprecipitation (ChIP) - qPCR

The ChIP assay was used to confirm the binding capacity between E2F3 and the rs11676272 region in HEK293T cells. In brief, the cells were fixed with 1% formaldehyde solution at room temperature (RT) for 10 min. The reaction was quenched by adding 2 mL 10x glycine buffer for 5 min. The cells were washed twice with cold PBS and collected with 2 mL of 1x PBS containing protease inhibitors. Micrococcal nuclease (0.5 μL per $4 \times 10^6$ cells) was added and incubated at 37 °C for 20 min. Agarose gel electrophoresis was used to determine DNA fragment size. The IgG (CST, Cat. No. 2729) and E2F3 (Proteintech, 27615-1-AP)

were designated as the control and experimental groups. A total of 8 μg of crosslinked chromatin fragments and the corresponding antibodies were added to each IP reaction, which was then incubated at 4 °C overnight on a rotator. ChIP-level protein G magnetic beads (30 μL) were added to each immunoprecipitation reaction and incubated at 4 °C for 2 h. The enriched products were obtained by separating and rinsing the magnetic beads. The binding ability of E2F3 was determined by qPCR. The BeyoChIP™ Enzymatic ChIP Assay Kit (Beyotime, Cat. No. P2083S) was used for the ChIP assay according to the manufacturer's instructions.

## High-fat diet-induced obesity

Mice were housed in plastic cages with ad libitum diet and maintained at 22 °C with a 12-h dark/12-h light cycle. The standard chow diet was obtained from Keao Xieli Feed, and the high-fat diet was purchased from Research Diets (D12492, 60% fat calories). Mouse weight was monitored weekly throughout the experiments, and food consumption was detected for 7 consecutive days following the development of an obese phenotype.

## Metabolic parameter measurements

hADCY3 and Mut-hADCY3 mice were acclimated for 24 h in a comprehensive laboratory animal monitoring system (TSE systems, Bad Homburg, Germany). Following adaptation, metabolic parameters, including $O_2$ consumption, $CO_2$ production, and total locomotor activity, were continuously recorded over the subsequent 5 min according to the manufacturer's protocol. The respiratory exchange ratio (RER) was calculated as the $VCO_2$ to $VO_2$ ratio.

## Assessment of hypothalamic neuronal activity

The hADCY3 and Mut-hADCY3 mice were fasted for 24 h and then re-fed with an HFD. Two hours later, mice were euthanized, and brains were processed for c-Fos immunofluorescence staining. Neuronal activation was evaluated by quantifying c-Fos-positive cells using a confocal microscope.

## Immunofluorescence staining and image analysis

Mice were transcardially perfused with 4% paraformaldehyde (PFA). Brains were post-fixed in 4% PFA at 4 °C for 12–16 h, cryoprotected in 30% sucrose at 4 °C for 48 h, embedded in OCT compound, and coronally sectioned at a thickness of 40 μm on a cryostat (Leica). The MEF cells derived from hADCY3 and Mut-hADCY3 mice were seeded on coverslips in 24-well plates. Brain

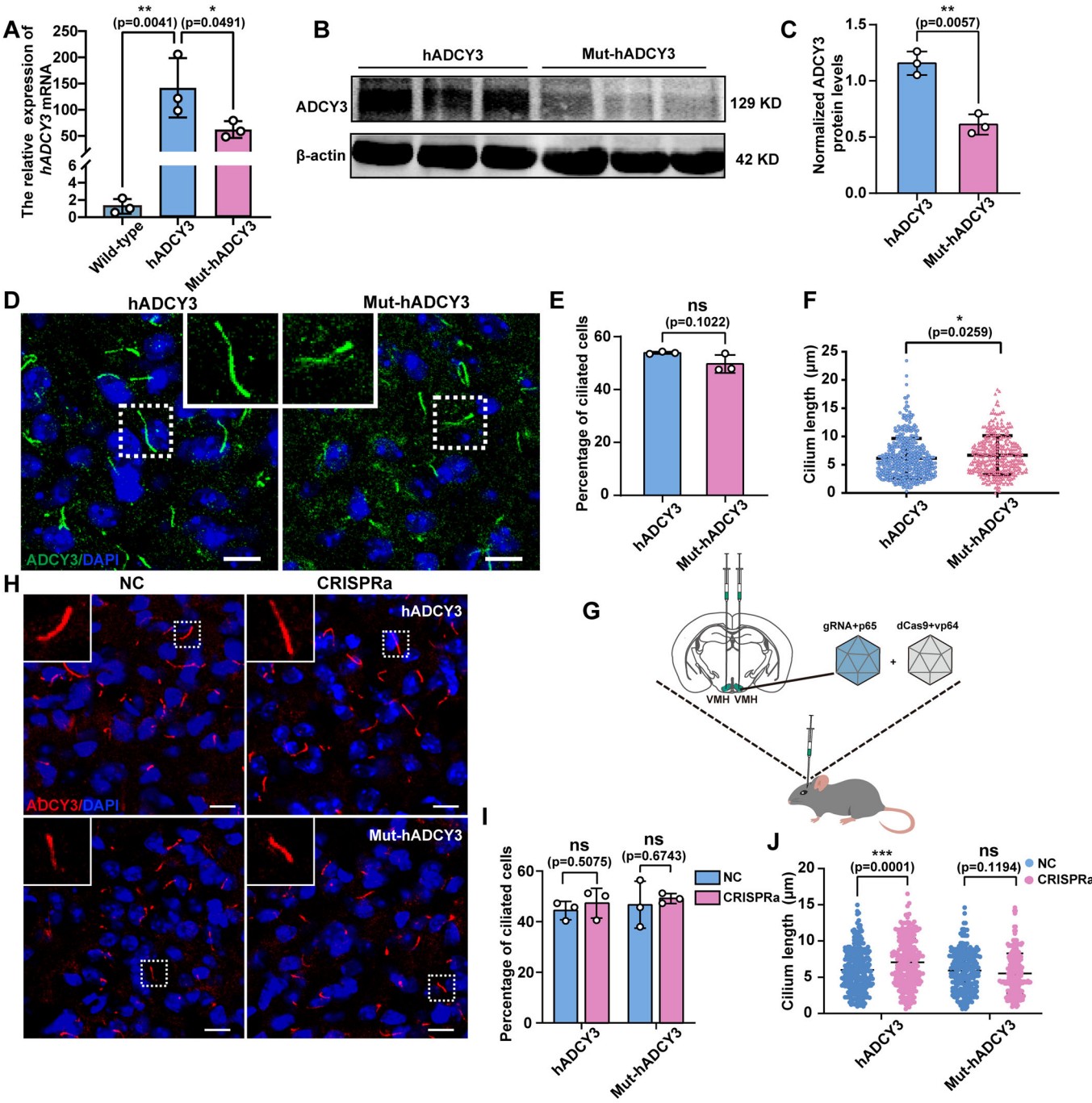

**Figure 6.  Mice with the rs11676272-C risk variant of hADCY3 result in lower ADCY3 levels and shorter cilia in vivo.**

(A) qPCR detecting the *hADCY3* gene (n = 3) expression levels in Wild-type, hADCY3 and Mut-hADCY3 mice. n = 3 groups from three independent experiments. (B, C) Western blot probing with anti-ADCY3 of the hypothalamus from the hADCY3 and Mut-hADCY3 mice. β-actin was used as a loading control. n = 3 groups from three independent experiments. (D–F) Cells in the VMH from the hADCY3 and Mut-hADCY3 mice were stained with anti-ADCY3 (green) antibody. DAPI (blue) was used to stain nuclei. Scale bars, 10 μm (D). Percentage of ciliated cells (E), and quantifications of cilium length (F). n = 3, a minimum of 200 cells were used for each group from three independent experiments. (G) Schematics of the stereotactic injection of the VMH (green circle). (H–J) Cells in the VMH from the hADCY3 and Mut-hADCY3 mice following injections of the NC or CRISPRa constructs were stained with anti-ADCY3 (red) antibody. DAPI (blue) was used to stain nuclei. Scale bars, 10 μm (H). Percentage of ciliated cells in the VMH from the hADCY3 and Mut-hADCY3 mice (I), and quantification of cilium length in the VMH from the hADCY3 and Mut-hADCY3 mice (J). n = 3 groups from three independent experiments. All data were presented as mean ± SD; *p < 0.05; **p < 0.01, ***p < 0.001, ns no significance. p values were calculated using one-way ANOVA in (A) and Student's t-test in (B–J). Source data are available online for this figure.

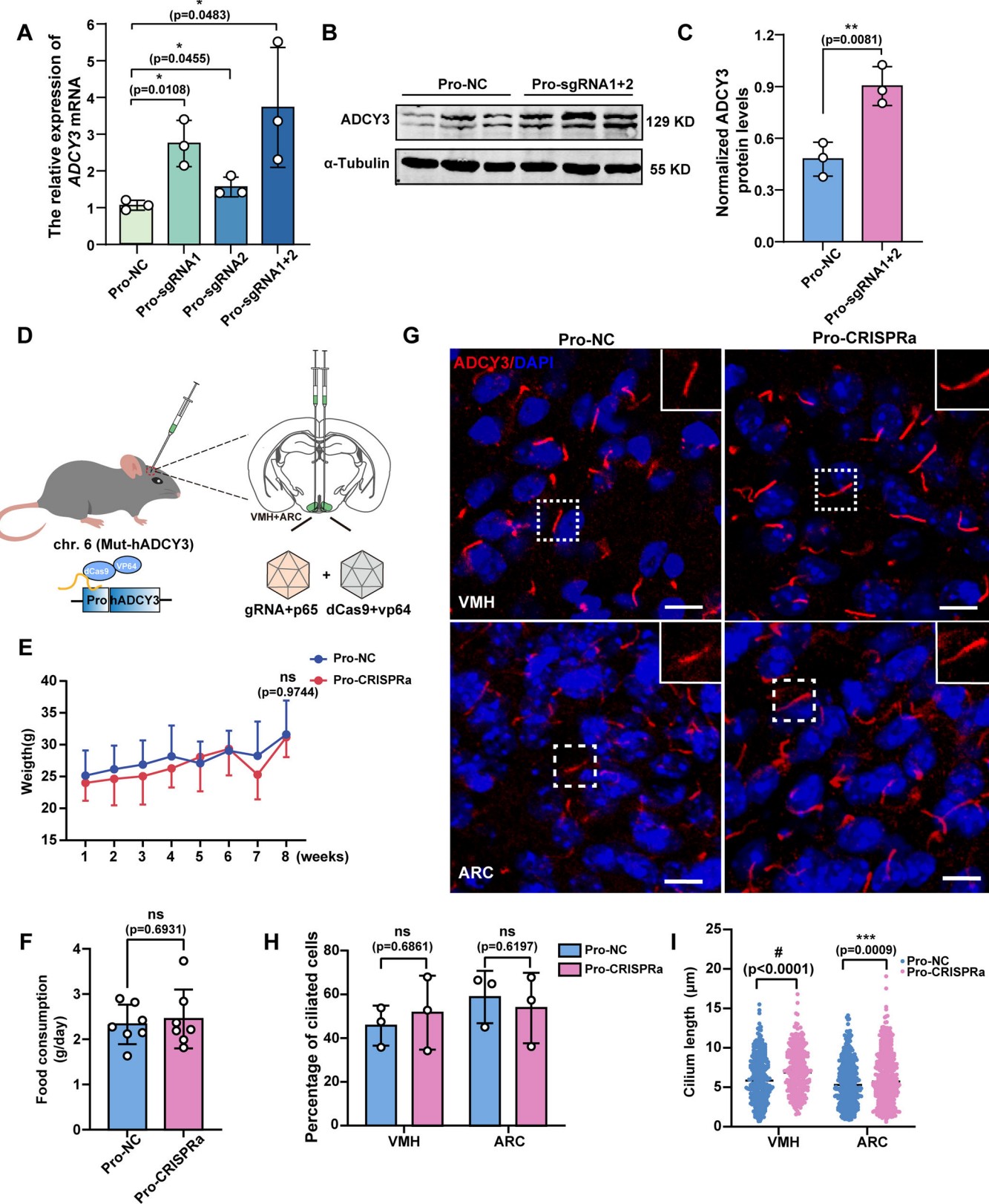

◀ **Figure 7.  Upregulation of hADCY3 expression level in the VMH and ARC partially rescues the shortened cilia in Mut-hADCY3 mice.**

(A) qPCR analysis of ADCY3 expressions in rs11676272-C HEK293T cells transfected with CRISPRa sgRNAs targeting the *hADCY3* promoter (Pro-sgRNA1, Pro-sgRNA2, and Pro-sgRNA1 + 2). $n = 3$ groups from three independent experiments. (B, C) Western blot probing with anti-ADCY3 in rs11676272-C HEK293T cells transfected with Pro-NC or Pro-sgRNA1 + 2 along with CRISPRa construct. α-Tubulin was used as a loading control. $n = 3$ groups from three independent experiments. (D) Schematics of the stereotactic injection of the VMH and ARC (green circle). (E) Quantification of body weights of the Mut-hADCY3 mice injected with Pro-NC or Pro-CRISPRa virus under HFD. $n = 6$ groups from three independent experiments. (F) Quantification of food consumption in Pro-NC and Pro-CRISPRa mice under HFD ($n = 7$). (G–I) Cells in the VMH and ARC from the Mut-hADCY3 mice following injections of the pro-NC or pro-CRISPRa virus were stained with anti-ADCY3 (red) antibody. DAPI (blue) was used to stain nuclei. Scale bars, 10 μm (G). Percentage of ciliated cells in the VMH and ARC from the Mut-hADCY3 mice (H) and quantification of cilium length in the VMH and ARC from the Mut-hADCY3 mice (I). $n = 3$ groups from three independent experiments. All data were presented as mean ± SD, *$p < 0.05$, **$p < 0.01$, ***$p < 0.001$, #$p < 0.0001$, ns no significance. $p$ values were calculated using one-way ANOVA in (A, E) and by Student's $t$-test in (C, F, H, I). Source data are available online for this figure.

sections and MEF cells were fixed with 4% PFA, permeabilized with 0.5% Triton X-100 in PBS for 15 min. After blocking in 10% goat serum in PBS, samples were incubated overnight at 4 °C with primary antibodies: rabbit anti-ADCY3 (Novus, NBP1-92683, 1:1000 dilution) and rabbit anti-ARL13B (Proteintech,17711-1-AP, 1:1000 dilution). After washing, samples were incubated with Alexa Fluor488- and 594-conjugated secondary antibodies (Thermo Scientific, 1:1000 dilution) for 2 h at room temperature, and then mounted with an anti-quenching mounting agent (Southern Biotech, 0100-01). Images were acquired using an Olympus FLUOVIEW FV3000 confocal microscope with a 40 × NA (brain sections) or 60 × NA (MEF cells) 1.4 Pan Apo oil immersion objective and processed with FV10-ASW software (Olympus). Primary cilium length was measured in ImageJ by importing immunofluorescence images with scale bars, followed by setting the scale via Analyze > Set Scale, individual cilia were manually traced, and cilium length was recorded using the measurement tool.

## Paraffin sections and H&E staining

Mice epididymal white adipose tissue and interscapular brown adipocytes were excised with dissecting scissors and were fixed in 4% PFA. Paraffin sections (5-μm thick) were prepared and deparaffinized for histologic analysis. H&E staining (Servicebio, G1005-1) was performed, and the images were acquired using the Olympus microscope with a bright-field imaging setup.

## Oil Red O staining

Fresh liver samples were snap-frozen in liquid nitrogen and then embedded in Tissue-Plus OCT Compound (Thermo Fisher Scientific) for frozen sectioning. Frozen sections (6-μm thick) were washed with 60% isopropanol for 2 min and stained with a 60% Oil Red O solution for 8 min. Following three washes with 60% isopropanol and distilled water, the nuclei were counterstained with hematoxylin. Finally, the slides were mounted for microscopy.

## RNA extraction and qPCR quantification

Total RNA was extracted using the RNAiso Plus (TAKARA, 9109), and first-strand cDNA was synthesized using HiScript III RT SuperMix with gDNA wiper (Vazyme, R323-01). The qPCR was performed with Taq Pro Universal SYBR qPCR Master Mix (Vazyme, Q712-02) on a QuantStudio Real-Time PCR system.

Gene expression levels were normalized to *GAPDH* levels and were determined by the $2^{-\Delta\Delta CT}$ method.

## Western blot

The samples were lysed in lysis buffer (P0013, Beyotime) with a protease inhibitor cocktail (100 x) (5871, Cell Signaling Technology) for 20 min on ice, followed by centrifugation at 12,000×$g$ for 20 min at 4 °C. The protein lysates were separated on 7.5% or 10% SDS-PAGE gels and transferred to PVDF membranes. The membrane was blocked for 1 h in TBS containing 5% non-fat milk, followed by incubation at 4 °C overnight or at RT for 2–3 h with primary antibodies against ADCY3 (Bioss, bs-2027R, 1:500 dilution; Invitrogen, PA5-35382,1:1000 dilution), α-Tubulin (Sigma, AC035, 1:5000 dilution), E2F3 (Proteintech, 27615-1-AP, 1:1500 dilution), GAPDH (TransGen Biotech, HC301-01, 1:1000 dilution), β-action (TransGen Biotech, HC201-01, 1:1000 dilution). Secondary antibodies conjugated with 800 nm fluorophores (SeraCare KPL, 1:10,000) were used at RT for 1 –2 h. Images were obtained using Odyssey software (Gene).

All oligos used in this paper are listed in Table EV2.

## Statistical analysis

All statistical analyses were performed using GraphPad Prism 9 software. The data are shown as mean ± SD. "*n*" represents the number of biologically independent samples or experiments. Details of the statistical analysis for each experiment can be found in the corresponding figure legends. All statistical analyses were calculated using an unpaired two-tailed Student's *t*-test unless otherwise mentioned. Data from more than two groups were calculated using one-way ANOVA and Bonferroni pairwise comparisons, as indicated in the figure legend (Ghosh and Bouchard 2017; Wu et al, 2016).

## Data availability

The off-target sequencing data are publicly accessible at: https://github.com/Shannon-Dong/EMBOR-2025-62007V1.

The source data of this paper are collected in the following database record: biostudies:S-SCDT-10_1038-S44319-026-00758-9.

## Peer review information

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

## Acknowledgements

We thank the Microscopy core facilities for technical assistance with imaging. This work was supported by the funding from the National Natural Science Foundation of China to Z.W. (32470645 and 32070567) and XH (32202840), Institute of Life Science and Green Development, Hebei University (050001-5000019), the Natural Science Foundation of Hebei Province of China to XH (C2023201032), and Priority-Funded Postdoctoral Research Project, Zhejiang Province to W.W. (ZJ2025118).

## Author contributions

**Weina Wang**: Conceptualization; Resources; Data curation; Software; Formal analysis; Funding acquisition; Validation; Visualization; Methodology; Writing—original draft; Writing—review and editing. **Yue Li**: Data curation; Methodology. **Sheng Dong**: Data curation; Software; Formal analysis; Writing—original draft. **Yuwei Liu**: Data curation; Validation; Methodology. **Chenghang Guo**: Data curation; Formal analysis; Methodology. **Yuzhe Su**: Data curation; Formal analysis; Methodology. **Wei Tian**: Data curation; Validation; Methodology. **Xiaoyu Hu**: Supervision; Funding acquisition; Investigation; Writing—original draft; Project administration; Writing—review and editing. **Zhenshan Wang**: Conceptualization; Supervision; Funding acquisition; Investigation; Methodology; Writing—original draft; Project administration; Writing—review and editing.

Source data underlying figure panels in this paper may have individual authorship assigned. Where available, figure panel/source data authorship is listed in the following database record: biostudies:S-SCDT-10_1038-S44319-026-00758-9.

## Disclosure and competing interests statement

The authors declare no competing interests.

# Expanded View Figures

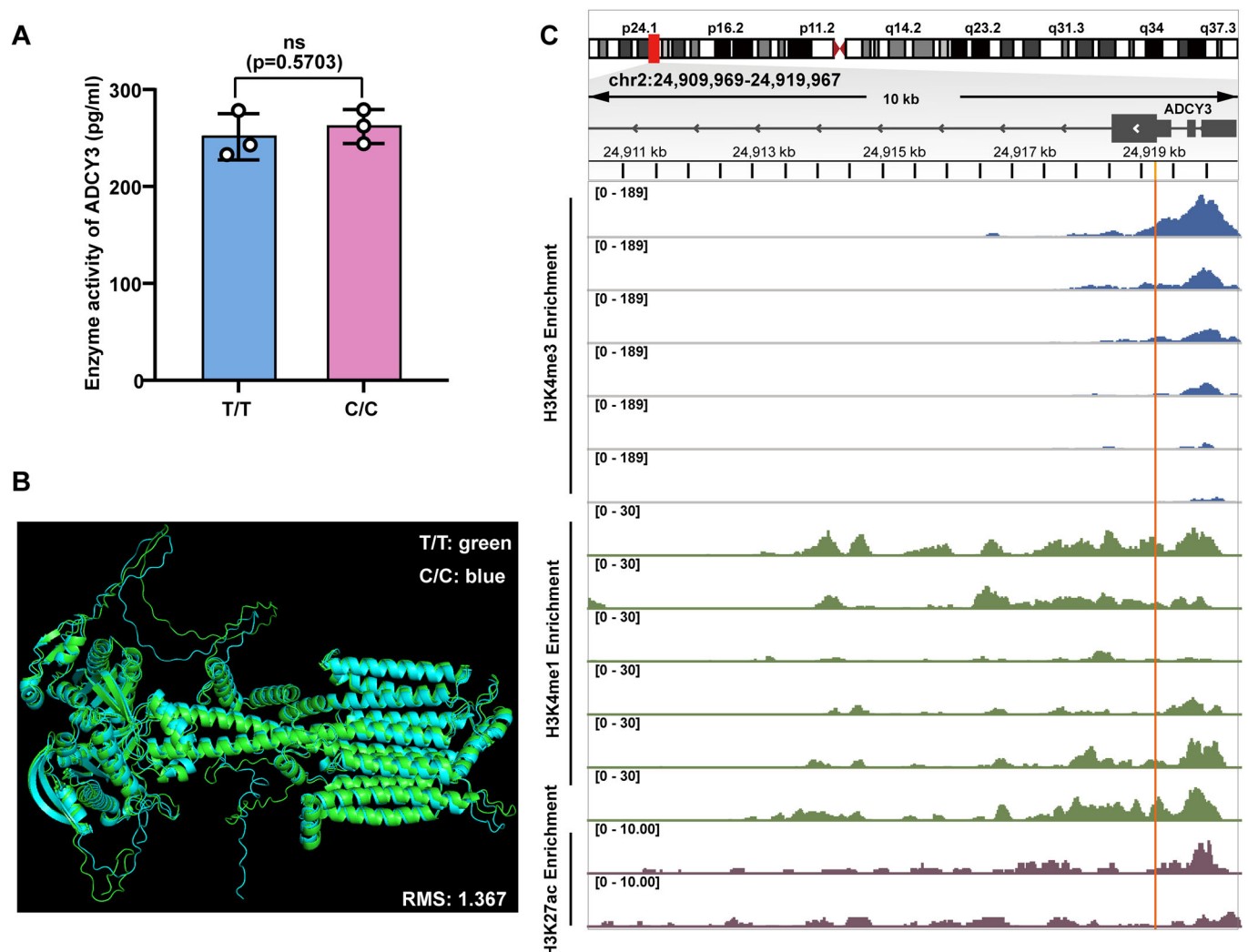

**Figure EV1. Characterization of ADCY3 enzymatic activity, protein structure, and locus epigenetics in the context of the rs11676272 variant.**

(A) ADCY3 enzymatic activity in T/T and C/C cells. $n = 3$ groups from three independent experiments. (B) AlphaFold predicted structural models of human ADCY3 carrying rs11676272-T and -C alleles. (C) Epigenetic annotations of H3K27ac, H3K4me1, and H3K4me3 marks at the rs11676272 locus in adipose tissue from the ENCODE database. rs11676272 is indicated by the orange vertical line. Data were presented as mean ± SD, ns not significant by Student's *t*-test.

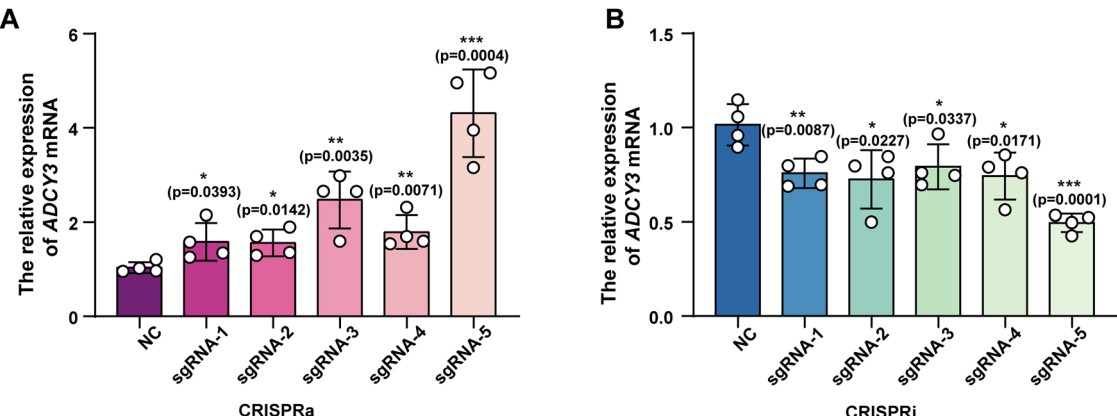

Figure EV2.  Validation of the sgRNA efficiency for CRISPR editing.

(A) qPCR analysis of *ADCY3* activation using indicated sgRNA sequences. (B) qPCR analysis of *ADCY3* inhibition using indicated sgRNA sequences. *GAPDH* was used as internal reference; $n = 4$ groups from three independent experiments; All data were presented as mean ± SD; *$p < 0.05$; **$p < 0.01$, ***$p < 0.001$ by one-way ANOVA.

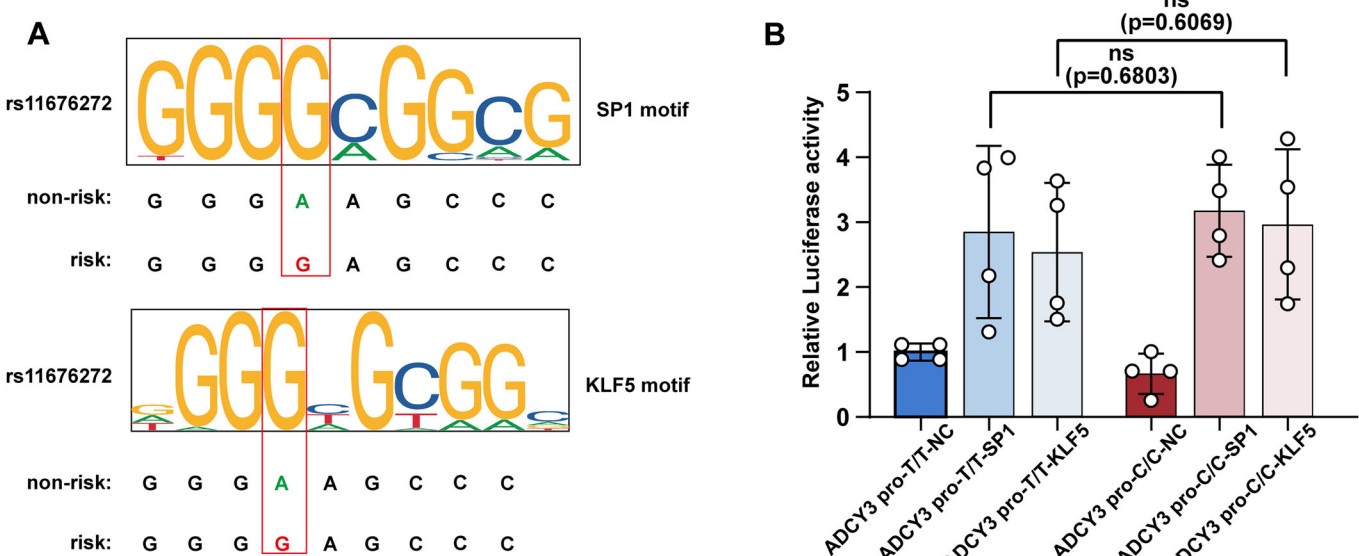

**Figure EV3. rs11676272-T or C alleles do not affect the binding affinity of SP1 and KLF5.**

(A) The SP1 binding motif (top) and KLF5 binding motif (bottom) and the rs11676272-C risk allele (complementary) and the rs11676272-A nonrisk allele (complementary) are shown. (B) Luciferase reporter assay performed after transient cotransfection of NIH3T3 cells with the indicated constructs. The DNA fragment containing either the T or C allele of the rs11676272 was cloned into the *ADCY3* promoter-driven luciferase reporter construct, which was cotransfected with either an empty vector or SP1 or KLF5-overexpressing vector. $n = 4$ groups from three independent experiments; All data were presented as mean ± SD, ns not significant by one-way ANOVA.

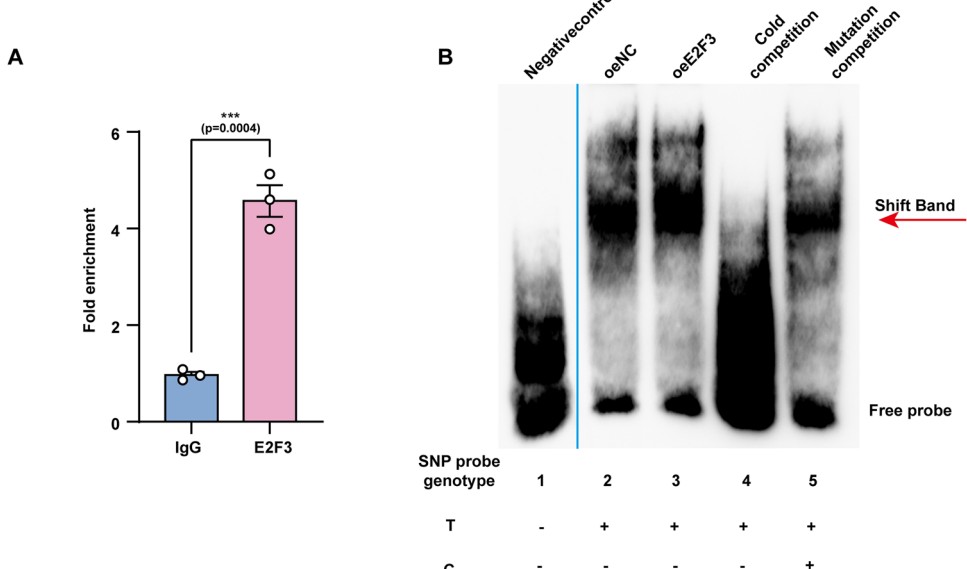

**A**

**B**

**Figure EV4. rs11676272-C allele specifically disrupts E2F3 binding capacity, whereas the -T allele preserves robust E2F3 recruitment.**

(A) ChIP-qPCR for *E2F3* at the DNA fragment containing rs11676272 region in HEK293T cells. $n = 3$ groups from three independent experiments. (B) EMSA showing differential affinities of the rs11676272-T nonrisk and -C risk alleles for E2F3. The red arrow indicates the shifted bands. Nuclear extracts were obtained from HEK293T cells transfected with either an empty vector or an E2F3-overexpressing construct. All data were presented as mean ± SD, ***$p < 0.001$ by Student's *t*-test. Source data are available online for this figure.

**A**

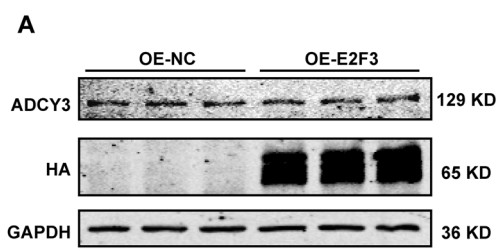

**B**

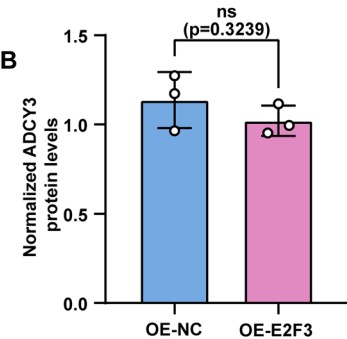

**Figure EV5. E2F3 does not bind to the rs11676272-C risk allele and enhance the ADCY3 expression.**

(A, B) Western blot showing E2F3 and ADCY3 expression levels in E2F3 overexpressing rs11676272-C/C cells. Cell lysates were obtained from HEK293T cells transfected with either an empty vector or an E2F3-overexpressing construct. GAPDH was used as a loading control. $n = 3$ groups from three independent experiments; All data were presented as mean ± SD, ns, not significant by Student's *t*-test.

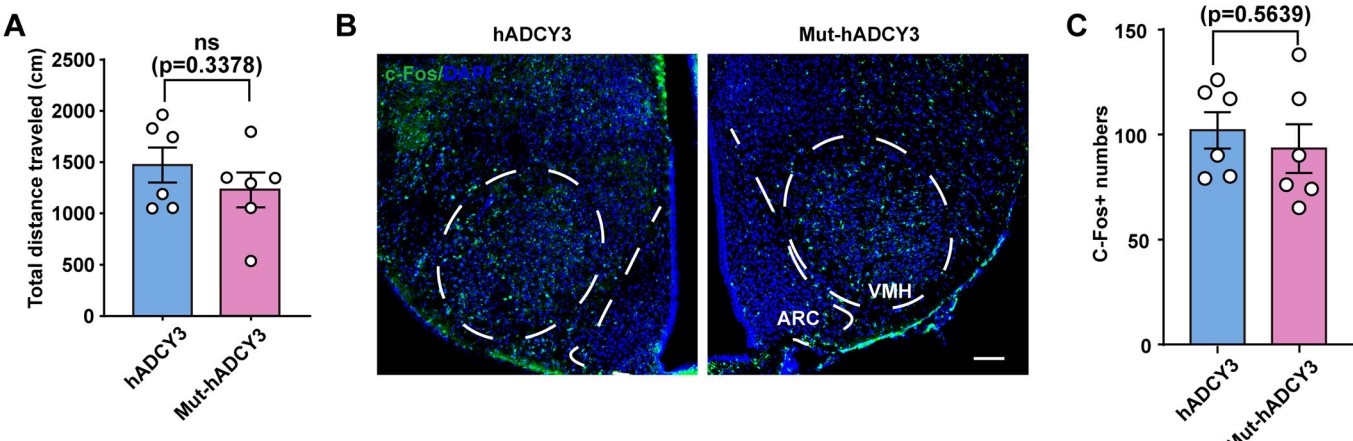

**Figure EV6. The total locomotor activity and neuronal activation of hADCY3 and Mut-hADCY3 mice showed no significant differences.**

(A) Total locomotor activity in hADCY3 and Mut-hADCY3 mice. $n = 6$ groups from three independent experiments. (B, C) Representative images (B) and quantification (C) of c-Fos immunofluorescence in VMH and ARC of the hADCY3 and Mut-hADCY3 mice. DAPI (blue) was used to stain nuclei. Scale bars, 100 μm. $n = 6$ groups from three independent experiments; All data were presented as mean ± SD; ns no significance by Student's $t$-test.

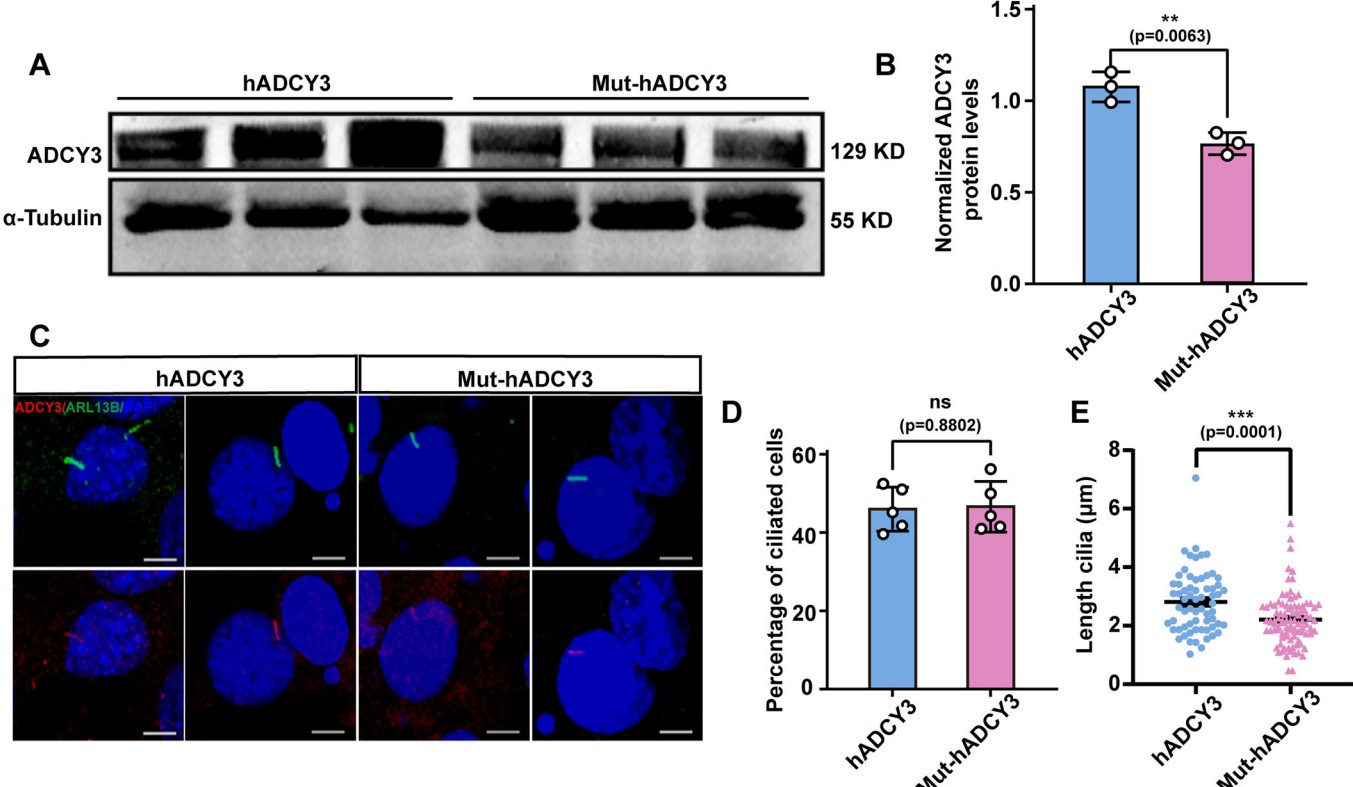

**Figure EV7. The hADCY3 mice with the rs11676272-C risk variant result in lower ADCY3 levels and shorter cilia in the MEF cells.**

(**A, B**) Western blot probing with anti-ADCY3 in MEF cells derived from hADCY3 and Mut-hADCY3 mice. α-Tubulin was used as a loading control. $n = 3$ groups from three independent experiments. (**C–E**) MEF cells from hADCY3 and Mut-hADCY3 mice were starved for 24 h, followed by staining with anti-ARL13B (green) and anti-ADCY3 (red) antibodies. DAPI (blue) was used to stain nuclei. Scale bars, 5 μm (**C**). Percentage of ciliated cells (**D**) and quantitative analysis of the cilium length (**E**) of MEF cells from hADCY3 and Mut-hADCY3 mice. $n = 3$ groups from three independent experiments; All data were presented as mean ± SD; **$p < 0.01$, ***$p < 0.001$, ns no significance by Student's *t*-test.

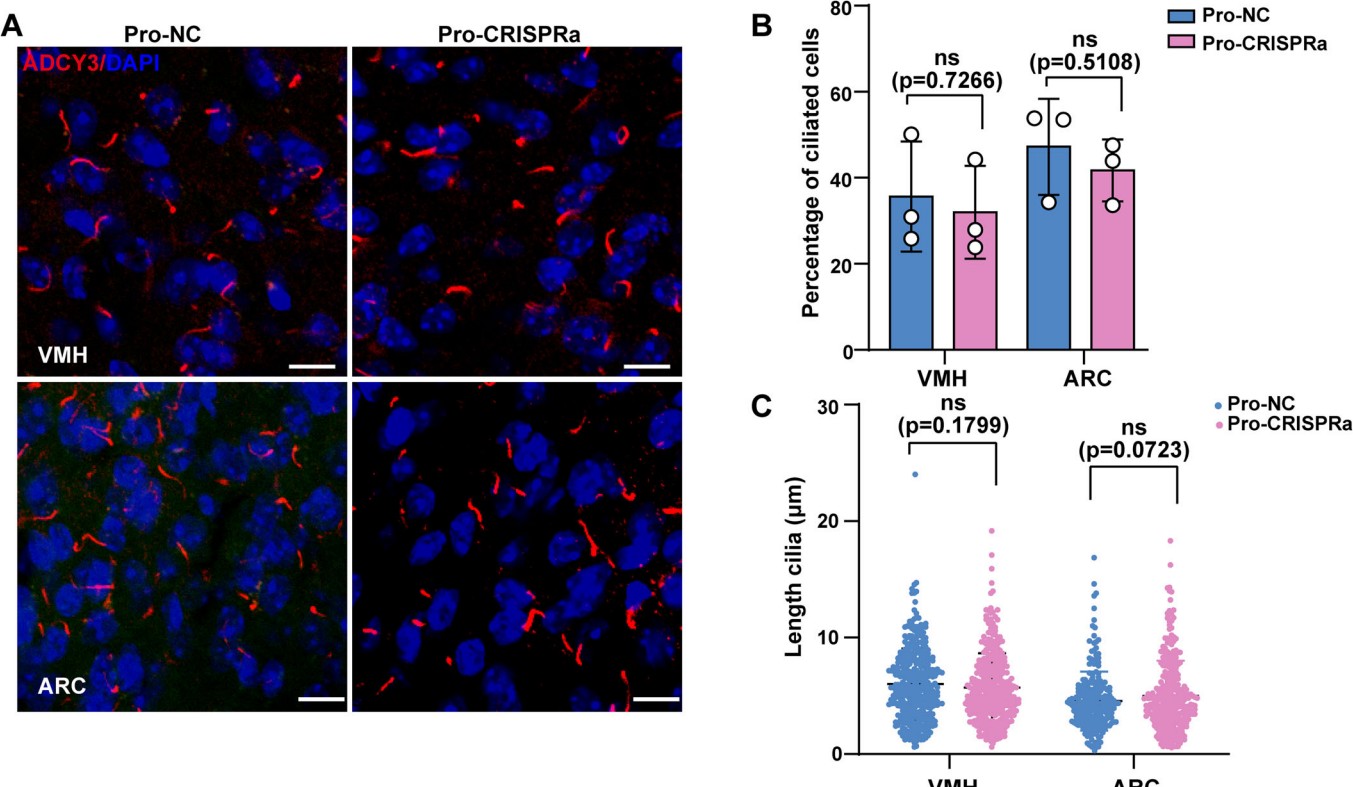

**Figure EV8. CRISPRa-AAV injection in the VMH and ARC did not affect the length of cilia in wild-type mice.**

(A) Cells in the VMH and ARC from the wild-type mice following injections of the pro-NC or pro-CRISPRa constructs were stained with anti-ADCY3 (red) antibody. DAPI (blue) was used to stain nuclei. Scale bars, 10 μm. (B, C) Percentage of ciliated cells in the VMH and ARC from the wild-type mice (B), and quantification of cilium length in the VMH and ARC from the wild-type mice (C). $n = 3$ groups from three independent experiments; The data were displayed as mean ± SD. ns no significance by one-way ANOVA.

