## [Peer Review File · EMBO Reports]

The exonic SNP rs11676272-C risk allele mediates diet-induced obesity and reduces enhancer activation

Xiaoyu Hu, Weina Wang, Yue Li, Sheng Dong, Yuwei Liu, Chenghang Guo, Yuzhe Su, Wei Tian, and Zhenshan Wang

Corresponding author(s): Xiaoyu Hu (xiaoyu.hu@hbu.edu.cn) , Zhenshan Wang (zswang@hbu.edu.cn)

Review Timeline:

Submission Date:	28th May 25
Editorial Decision:	11th Jul 25
Revision Received:	20th Nov 25
Editorial Decision:	11th Feb 26
Revision Received:	15th Feb 26
Accepted:	18th Mar 26

Editor: Esther Schnapp

Transaction Report:

Dear Dr. Hu,

Thank you for your patience while your manuscript was peer-reviewed at EMBO reports. We have now received the full set of referee reports as well as cross-comments from referee 3; all pasted below.

As you will see, the referees acknowledge that the findings are potentially interesting. However, together they also raise quite a number of concerns and it is clear that your study is a very borderline case. I asked all referees for cross-comments and what a minimal set of revisions would entail but unfortunately, only referee 3 got back to us. However, we agree with this referee, that 2 points must be addressed for a revision at EMBO reports: The enzymatic function of mutant *Adcy3* must be assessed and the KI mutant mice must be (much) better characterized. A role for cilia in the obese phenotype will most likely be too difficult to further elucidate within a 3-4 month revision timeframe. All other concerns must be addressed too.

If you decide to embark on such revisions, I would like to invite you to revise your manuscript with the understanding that the referee concerns must be fully addressed and their suggestions taken on board. Please address all referee concerns in a complete point-by-point response. Acceptance of the manuscript will depend on a positive outcome of a second round of review. It is EMBO reports policy to allow a single round of major revision only and acceptance or rejection of the manuscript will therefore depend on the completeness of your responses included in the next, final version of the manuscript.

We realize that it is difficult to revise to a specific deadline. In the interest of protecting the conceptual advance provided by the work, we recommend a revision within 3 months (11th Oct 2025). Please discuss the revision progress ahead of this time with the editor if you require more time to complete the revisions.

- 1) A data availability section providing access to data deposited in public databases is missing. If you have not deposited any data, please add a sentence to the data availability section that explains that.
- 2) Your manuscript contains statistics and error bars based on $n=2$. Please use scatter blots in these cases. No statistics should be calculated if $n=2$.

3) We replaced Supplementary Information with Expanded View (EV) Figures and Tables that are collapsible/expandable online. A maximum of 5 EV Figures can be typeset. EV Figures should be cited as 'Figure EV1, Figure EV2' etc... in the text and their respective legends should be included in the main text after the legends of regular figures.

5) a complete author checklist, which you can download from our author guidelines <https://www.embopress.org/page/journal/14693178/authorguide>. Please insert information in the checklist that is also reflected in the manuscript. The completed author checklist will also be part of the RPF.

6) Please note that all corresponding authors are required to supply an ORCID ID for their name upon submission of a revised manuscript (<https://orcid.org/>). Please find instructions on how to link your ORCID ID to your account in our manuscript tracking system in our Author guidelines

<<https://www.embopress.org/page/journal/14693178/authorguide#authorshippinguidelines>>

12) All Materials and Methods need to be described in the main text using our 'Structured Methods' format, which is required for all research articles. According to this format, the Methods section includes a separate Reagents and Tools Table file (listing key reagents, experimental models, software and relevant equipment and including their sources and relevant identifiers) and a Methods and Protocols section describing the methods using a step-by-step protocol format. The aim is to facilitate adoption of the methodologies across labs. More information on how to adhere to this format as well as a downloadable template (.docx) for the Reagents and Tools Table can be found in our author guidelines:
<https://www.embopress.org/page/journal/14693178/authorguide#structuredmethods>.

An example of a Method paper with Structured Methods can be found here: <https://www.embopress.org/doi/full/10.1038/s44320-024-00037-6#sec-4>

I look forward to seeing a revised form of your manuscript when it is ready.

Best regards,
Esther,

Referee #1:

in GWAS database, ADCY3 SNPs have been linked to obesity. In a large-scale cohort study of 50,000 children, the author found that, among 20 candidate loci, rs11676272 locus located at Exon 1 of ADCY3 gene to be most statistically significant. rs11676272 T>C is previously known to be associated with obesity, potentially to decrease its enzyme activity (T>C leads to substitution of aa 107 from serine to proline).

The authors' postulation was that this region with histone modifications and ATAC-peaks may function as an enhancer to affect ADCY3 transcription. The authors performing luciferase reporter- promoter assays using TT or CC mutated constructs in various cell lines. They also employed CRISPRi, CRISPRa as well as mutation of T to C in HEK293 cells to demonstrate this region functioning as an enhancer. The authors detected a ETF3 binding site at this region and the T to C mutation caused lower ETF3 binding by ChIP assays. CRISPR deletion of ETF3 binding site caused lowering of ADCY3 expression and rs11676272 T>C could not increase ADCY3 expression upon ETF3 overexpression. Finally, the authors also generated KI mice for WT human ADCY3 and T to C mutation to show higher obesity under HFD feeding. They also showed shorter cilia in MEFs from T to C mutation KI mice, and delivery of CRISPRa AAV into VMH increased cilia length, but did not rescue obesity phenotype. The authors performed extensive studies of the region in exon1 containing rs11676272 as an enhancer and effect of rs11676272 T>C mutation. However, a major concern is that potential changes in enzyme activity of this missense mutation were not addressed at all. Thorough characterization of adenylyl cyclase as an enzyme as affected by serine to proline substitution of aa 107 position (rs11676272) needs to be performed first, before studying whether this region functions as an enhancer to affect ADCY3 expression levels. Moreover, although the authors showed obesity phenotype of whole body rs11676272 T>C KI mice, with the observed changes in food intake, this may not be from direct effect on adipose tissue. The authors did not show how this mutation causes obesity. Other points to be addressed are as listed below.

1. If this region acts as an enhancer, the authors need to document changes in mRNA levels. However, only protein levels in various experiments, such as Figures 2, 3 and 6, were shown.
2. The cAMP levels in cells having C/C mutated version in the absence of forskolin or IBMX treatment appear to be higher in Fig. 2J, but the authors did not explain this result.
3. The authors did not determine ADCY3 expression in hADCY KI mice, which would be critical in demonstrating this substitution altering its putative enhancer function and not altering enzyme activity of ADCY with serine to proline substitution, *in vivo*.
4. Increased adiposity upon HFD feeding of whole body hADCY with mutation KI mice could be from increased food intake as shown in Fig. 5D, not from the direct effect of ADCY3 on adipose tissue. Overall, the authors did not address how WT and T to C substituted hADCY3 affected WAT and BAT resulting in obesity.
5. It is also unclear why food intake of mice was affected only when on HFD but not on chow diet (Fig. 5D). Similarly, the body weights were affected only on HFD feeding of WT hADCY3 KI mice (Fig. 5C).
6. Fig. 5 showing tissue sections of WAT and BAT of human WT KI mice are questionable as adipocyte size of WAT and BAT are similar, when adipocytes of BAT are usually smaller than those in WAT. Moreover, adipocytes of BAT are known to be multi-locular, when those in WAT are uni-locular. The tissue sections of BAT appear to be uni-locular, and adipocyte size of WAT and BAT appear to be similar in WT ADCY3 KI mice (Fig. 5F).
7. The ADCY3 function in cilium development as affected by the substituted hADCY3 in mice did not affect body weight and did not rescue obesity of hADCY3 substitution KI mice, showing this observation is unrelated to above studied obesity phenotype and this is a new topic. In this regard, however, if changes in cilia length from ADCY3 at the VMH affects VMH function, changes in food intake could be predicted, but the authors did not examine food intake, only inferred by body weight.

Referee #2:

In the present article, the authors aim to investigate the effect of the exonic SNP rs11676272 (located in ADCY3 gene) on obesity. ADCY3 is a known gene involved in monogenic obesity and SNPs at this locus were found to be associated with polygenic adiposity. The authors report that this variant influences ADCY3 expression, potentially through enhancer activity.

However, the reviewer raises several major concerns regarding the interpretation of the results and the strength of the evidence supporting rs11676272 as the causal variant at this locus.

First, what is the correlation between rs11676272 and other SNPs in linkage disequilibrium (LD) in this region? In other words, what evidence supports rs11676272 as the true functional variant? Its association signal in obesity GWAS, while notable, is not sufficient on its own to demonstrate causality. Has any fine-mapping analysis been performed to prioritize this variant over others in LD? While RegulomeDB can provide suggestive regulatory annotations, it cannot be considered conclusive proof of functionality or causality in the context of GWAS loci.

Additionally, the choice of cell models used to assess the functional effects of the variant is problematic. HEK293T cells are not metabolically relevant, and ADCY3 is not highly expressed in kidney tissue, as confirmed by the GTEx portal. This significantly limits the biological relevance of the genome editing experiments performed in these cells. The same concern applies to the use of NIH3T3 cells, which are murine fibroblasts and not appropriate models for studying obesity-related mechanisms. Importantly, genetic and functional studies have shown that ADCY3 plays a key role in obesity through its activity in hypothalamic neurons and insulin-responsive tissues such as skeletal muscle.

Moreover, rs11676272 is a coding variant (p.Ser107Pro). It is therefore surprising that no experiments were performed to evaluate whether this missense substitution affects protein function, structure, or stability, prior to hypothesizing a regulatory role. This aspect is critical and is not sufficiently addressed in the manuscript.

The observed difference in luciferase activity shown in Figure 1 (approximately 1.4-fold increase in T/T compared to the reference allele in HEK cells) appears modest, raising questions about its biological significance and how such an effect could translate into phenotypic consequences in mouse models.

The authors state that "it is not possible to manipulate ADCY3 in humans, resulting in a paucity of experimental evidence that could substantiate direct causality". This is incorrect: several human brain cell lines and induced pluripotent stem cells (iPSCs) differentiated into hypothalamic or neuronal lineages are currently available and could provide more relevant systems to study ADCY3 function in the context of obesity.

Minor comments: The western blots shown in Figures 2 and 3 are not clean and lack sufficient resolution and contrast, which compromises the interpretation of the protein expression results.

Referee #3:

Type 3 adenylyl cyclase (encoded by ADCY3) is predominantly expressed in neuronal primary cilia throughout the brain and a key enzyme mediating the cAMP signaling in the ciliary microdomain. ADCY3 is strongly associated with obesity, supported by numerous lines of GWAS studies and experimental studies on mouse models. The submitted report studied the causal relationship of SNP rs11676272 (T > C) with obesity. It provides evidence showing that the rs11676272 SNP region very likely function as an eExon that preferentially binds to the transcription factor E2F3 to regulate AC3 expression, with the C risk allele decreasing the binding. There are multiple pieces of interesting data. The manuscript is well-written and figure panels nicely presented. The findings derived from molecular genetics study are solid and novel. The study also attempted to link down-regulated AC3 expression to ciliary function in the brain centers that regulate energy balance, claiming the reduced AC3 expression causes HFD-induce weight-gain by shortening primary cilia in the VMH. This claim is not well substantiated by presented data.

Major points:

- (1) The mechanistic evidence on obesity is not compelling, as the cilia shortening effect observed in the VMH of the C allele knock-in mice is weak.
- (2) The markedly reduced AC3 protein expression in the mut-hADCY3 hypothalamus compared to that of hADCY3 does not align with the relatively mild shortening of primary cilia.
- (3) The effects of C allele knock-in on locomotor activity, energy expenditure, and overall neuronal activity were not assessed.
- (4) The SNP rs11676272 C risk allele likely affects either AC3 activity or expression systemically. A major limitation of the study is its exclusive focus on the VMH. The link between reduced cilia length in the VMH and HFD-induced weight gain is weak. Alternative mechanisms should be explored.
- (5) The claim of a causal relationship, particularly regarding to obesity mechanism, appears to be an over-interpretation, as the underlying mechanisms of AC3 association with obesity remain unclear and ciliary defects-caused obesity can result from multiple contributing factors.

Minor points:

- (1) Abstract: "providing offering", please delete one.
- (2) Statistical analysis: A normality test should be performed to confirm that the data are normally distributed before applying Student's t-test. If normality cannot be assumed, the Mann-Whitney U test should be used for unpaired samples, and the Wilcoxon signed-rank test for paired samples.

Responses to Reviewers

We thank the three reviewers for their critical comments. Our replies are detailed under each specific comment below in blue, and we also labeled our modifications in red in the revised manuscript.

Referee #1:

In GWAS database, ADCY3 SNPs have been linked to obesity. In a large-scale cohort study of 50,000 children, the author found that, among 20 candidate loci, rs11676272 locus located at Exon 1 of ADCY3 gene to be most statistically significant. rs11676272 T>C is previously known to be associated with obesity, potentially to decrease its enzyme activity (T>C leads to substitution of aa 107 from serine to proline).

The authors' postulation was that this region with histone modifications and ATAC-peaks may function as an enhancer to affect ADCY3 transcription. The authors performing luciferase reporter- promoter assays using TT or CC mutated constructs in various cell lines. They also employed CRISPRi, CRISPRa as well as mutation of T to C in HEK293 cells to demonstrate this region functioning as an enhancer. The authors detected a ETF3 binding site at this region and the T to C mutation caused lower ETF3 binding by ChIP assays. CRISPR deletion of ETF3 binding site caused lowering of ADCY3 expression and rs11676272 T>C could not increase ADCY3 expression upon ETF3 overexpression. Finally, the authors also generated KI mice for WT human ADCY3 and T to C mutation to show higher obesity under HFD feeding. They also showed shorter cilia in MEFs from T to C mutation KI mice, and delivery of CRISPRa AAV into VMH increased cilia length, but did not rescue obesity phenotype.

Main points:

1. The authors performed extensive studies of the region in exon1 containing rs11676272 as an enhancer and effect of rs11676272 T>C mutation. However, a major concern is that potential changes in enzyme activity of this missense mutation were

not addressed at all. Thorough characterization of adenylyl cyclase as an enzyme as affected by serine to proline substitution of aa 107 position (rs11676272) needs to be performed first, before studying whether this region functions as an enhancer to affect ADCY3 expression levels.

We thank the reviewer for raising this important point. In response, we conducted the following experiments: First, we measured the enzymatic activity of ADCY3 and found no significant differences between the T/T and C/C variant (Fig. EV1A). We then attempted to purify protein *in vitro* to further analyze its enzymatic kinetics. Unfortunately, despite optimizing various purification conditions, the purified protein tended to aggregate, and we were unable to obtain ADCY3 protein (ADCY3-Flag MW: 130.3 KDa) samples with satisfactory activity (See the figure below). To further evaluate the impact of this variant on protein structure, we performed protein structure prediction using AlphaFold Server. The root-mean-square-deviation (RMSD) was 1.3678 (Fig. EV1B), indicating that the C/C variant protein structure exhibits minimal differences from the T/T structure and maintains good stability. Based on these results, we propose that rs11676272 variant does not affect ADCY3 enzymatic activity.

Fig. 1 Flag-tag affinity chromatography and size exclusion chromatography (SEC) purification. **A–B.** SDS-PAGE from the the first Flag-tag affinity chromatography (A) and SEC profile following affinity chromatography purification (B). **C–D.** SDS-PAGE from the second Flag-tag affinity chromatography (C) and SEC profile following affinity chromatography purification (D). Aggregation protein is represented by ①, and degradation protein is represented by ② in C and D.

2. Moreover, although the authors showed obesity phenotype of whole body rs11676272 T>C KI mice, with the observed changes in food intake, this may not be from direct effect on adipose tissue. The authors did not show how this mutation causes obesity.

We appreciate the reviewer's insightful comment concerning the mechanistic basis of the rs11676272 variant in obesity. Emerging research highlights that alterations in hypothalamic neuronal cilium length as a critical contributor to obesity. For instance, in rats, shortened cilia in MC4R-expressing hypothalamic neurons impair brown adipose tissue thermogenesis and energy expenditure, leading to obesity.(Oya *et al*, 2024) Another study demonstrated that specifically ciliary shortening in the mouse VMH is sufficient to induce an obese phenotype.(Yang *et al*, 2022) We therefore hypothesize that ciliary shortening represents an important factor in obesity pathogenesis. Nevertheless, we cannot exclude the involvement of other factors, such as altered feeding behavior and impaired ADCY3 signaling in adipose tissue. These potential interpretations have been elaborated in the Discussion section of our revised manuscript.

3. If this region acts as an enhancer, the authors need to document changes in mRNA levels. However, only protein levels in various experiments, such as Figures 2, 3 and 6, were shown.

We appreciate the reviewer's valuable suggestion. Accordingly, we have performed additional qPCR assays and included the revised mRNA expression data in Figures 2, 3, and 6. These results confirm transcriptional regulation and further strengthen the evidence that this region acts as an enhancer.

4. The cAMP levels in cells having C/C mutated version in the absence of forskolin or IBMX treatment appear to be higher in Fig. 2J, but the authors did not explain this result.

We appreciate the reviewer's attention to this detail. Although basal cAMP levels in C/C cells appear slightly elevated compared to those in T/T cells in Fig. 1I, we attribute this modest difference to inter-group variation, as it was not statistically significant. To validate this observation, we repeated cAMP measurements in C/C and T/T cells without forskolin or IBMX treatment. The results consistently showed no significant differences between groups, reinforcing the conclusion that the rs11676272 variant does not alter basal cAMP levels.

Fig.2 cAMP levels in cells carrying different rs11676272 alleles in the absence of forskolin and IBMX. n = 4 groups from three independent experiments. Data were presented as mean \pm SD, ns, no significance by Student's *t*-test.

5. The authors did not determine ADCY3 expression in hADCY KI mice, which would be critical in demonstrating this substitution altering its putative enhancer function and not altering enzyme activity of ADCY with serine to proline substitution, *in vivo*.

We thank the reviewer for this insightful comment. Accordingly, we have assessed ADCY3 expression levels in hADCY3-KI mice, and the corresponding data have been incorporated into the revised manuscript as Fig. 6A.

6. Increased adiposity upon HFD feeding of whole body hADCY with mutation KI mice could be from increased food intake as shown in Fig. 5D, not from the direct effect of ADCY3 on adipose tissue. Overall, the authors did not address how WT and

T to C substituted hADCY3 affected WAT and BAT resulting in obesity.

We are grateful to the reviewer for their insightful comments and apologize for any lack of clarity in our manuscript. Our data reveal that the rs11676272 variant leads to a comprehensive obesity-related phenotype in mice, including increased body weight and food intake, decreased respiratory quotient, shortened cilia, elevated lipid accumulation in white adipose tissue and liver, and reduced brown adipocyte size. We fully agree that the mechanistic pathway linking this variant to obesity is complex and not yet fully elucidated. Previous studies have shown that ADCY3 plays a crucial role in BAT, and the loss of its variant ADCY3-AT enhances energy expenditure and protects against obesity and metabolic disorders (Khani et al, 2024). Moreover, ADCY3 is the predominant adenylate cyclase isoform expressed in WAT, and its mutation markedly impairs adenylate cyclase activity in this tissue (Pitman et al, 2014). This constitutes a major focus of our future mechanistic investigations. An explanation of these considerations is provided in the Discussion section of the revised manuscript.

7. It is also unclear why food intake of mice was affected only when on HFD but not on chow diet (Fig. 5D). Similarly, the body weights were affected only on HFD feeding of WT hADCY3 KI mice (Fig. 5C).

We thank the reviewer for raising this important question. The observed diet-specific phenotype—manifested as differences in body weight and food intake under high-fat diet (HFD) but not chow diet (CD) conditions—likely arises from the interplay between genetic susceptibility and metabolic stress. We propose two primary explanations: First, the rs11676272 variant may confer increased sensitivity to HFD-induced metabolic challenge. Under CD feeding, metabolic homeostasis is largely preserved, rendering the variant's physiological effects subthreshold. In contrast, HFD exerts sustained pressure on energy-regulatory pathways, potentially unmasking the latent physiological consequences of this variant. Second, as discussed in the relevant section, the mutation may impair ciliary structure and function, leading to compromised nutrient sensing. This defect could alter the perception of response to

dietary lipids, thereby predisposing animals to hyperphagia and weight gain under HFD conditions. To address this, we have elaborated on these mechanisms in the Discussion section of the revised manuscript, including the implications for future studies.

8. Fig. 5 showing tissue sections of WAT and BAT of human WT KI mice are questionable as adipocyte size of WAT and BAT are similar, when adipocytes of BAT are usually smaller than those in WAT. Moreover, adipocytes of BAT are known to be multi-locular, when those in WAT are uni-locular. The tissue sections of BAT appear to be un i-locular, and adipocyte size of WAT and BAT appear to be similar in WT ADCY3 KI mice (Fig. 5F).

We thank and agree with reviewer for this comment. BAT volume also exhibited an increasing trend under high-fat diet conditions(Kang *et al*, 2022). To ensure the reliability and accuracy of our data, the experiments involving both BAT and WAT were repeated with an increased sample size. The results from these repeated experiments and subsequent statistical analysis have been updated in Fig. 5F. Furthermore, although BAT is typically characterized by multilocular lipid droplets under physiological conditions, in contrast to the unilocular morphology of WAT. HFD challenge can induce significant morphological remodeling in BAT, including lipid droplet enlargement and a reduction in multilocularity, as reported in the literature(Yu *et al*, 2024).

9. The ADCY3 function in cilium development as affected by the substituted hADCY3 in mice did not affect body weight and did not rescue obesity of hADCY3 substitution KI mice, showing this observation is unrelated to above studied obesity phenotype and this is a new topic. In this regard, however, if changes in cilia length from ADCY3 at the VMH affects VMH function, changes in food intake could be predicted, but the authors did not examine food intake, only inferred by body weight.

We thank the reviewer for these insightful comments. As shown in Fig. 7, our experiment aimed to rescue the obesity phenotype induced by the rs11676272 variant

using CRISPRa-mediated activation of the human *ADCY3* promoter. However, we were unable to reverse the obesity phenotype in Mut-hADCY3 mice, and supplementary food intake measurements showed no significant changes (Fig. 7F). Notably, the treatment significantly increased cilia length. Given that obesity itself shortens cilia (as shown in control obese mice), this confirms that the CRISPRa treatment produced the expected local effect. We attribute the failure to rescue the overall obesity phenotype to the systemic nature of the rs11676272 variant, while our CRISPRa-mediated activation was limited to specific local sites such as VMH. The scale and strength of this targeted approach may therefore be insufficient to counteract systemic metabolic disruptions. The updated results on food intake, along with the corresponding explanation, are detailed in lines 431–447 of the revised manuscript.

Referee #2:

In the present article, the authors aim to investigate the effect of the exonic SNP rs11676272 (located in *ADCY3* gene) on obesity. *ADCY3* is a known gene involved in monogenic obesity and SNPs at this locus were found to be associated with polygenic adiposity. The authors report that this variant influences *ADCY3* expression, potentially through enhancer activity.

However, the reviewer raises several major concerns regarding the interpretation of the results and the strength of the evidence supporting rs11676272 as the causal variant at this locus.

1. First, what is the correlation between rs11676272 and other SNPs in linkage disequilibrium (LD) in this region? In other words, what evidence supports rs11676272 as the true functional variant? Its association signal in obesity GWAS, while notable, is not sufficient on its own to demonstrate causality. Has any fine-mapping analysis been performed to prioritize this variant over others in LD? While RegulomeDB can provide suggestive regulatory annotations, it cannot be considered conclusive proof of functionality or causality in the context of GWAS loci.

We appreciate the reviewer's insightful comment on linkage disequilibrium and fine-

mapping. We agree that statistical association and computational annotations alone are insufficient to establish causality, which is precisely why we employed direct functional experimentation. Our approach was to move beyond correlation and test the functional impact of rs11676272 directly. By engineering models that differ only at this specific nucleotide, we demonstrate that the rs11676272-C risk allele is sufficient to reduce ADCY3 protein levels, alter ciliary length, and promote weight gain. This experimental evidence significantly elevates rs11676272 from a statistical association to a functionally validated variant. While it remains formally possible that other SNPs in LD are also functional, our data establish rs11676272 as a bona fide functional variant and a central player in this association signal.

2. Additionally, the choice of cell models used to assess the functional effects of the variant is problematic. HEK293T cells are not metabolically relevant, and ADCY3 is not highly expressed in kidney tissue, as confirmed by the GTEx portal. This significantly limits the biological relevance of the genome editing experiments performed in these cells. The same concern applies to the use of NIH3T3 cells, which are murine fibroblasts and not appropriate models for studying obesity-related mechanisms. Importantly, genetic and functional studies have shown that ADCY3 plays a key role in obesity through its activity in hypothalamic neurons and insulin-responsive tissues such as skeletal muscle.

The brain, particularly the hypothalamus, plays a central role in regulating energy homeostasis and metabolism, making neuronal cells a relevant model for studying metabolic disorders. As shown in Fig. 2D, the effect of the rs11676272 variant on the *ADCY3* promoter was consistent between HEK293T cells and neuronal SY5Y cells. However, due to the difficulty in genetically modifying SY5Y cells and their inability to form stable monoclonal populations precluded further mechanistic studies. Therefore, HEK293T cells served as the most suitable model for elucidating the rs11676272 regulatory mechanism. To ensure robustness, we replicated the original NIH3T3 cell experiments in HEK293T cells and have revised the main text accordingly (Fig. 3C, D; Fig. EV3B). The results were consistent between these two

cell types. We acknowledge the limitations of using HEK293T cells and have addressed this point in the Discussion section.

3. Moreover, rs11676272 is a coding variant (p.Ser107Pro). It is therefore surprising that no experiments were performed to evaluate whether this missense substitution affects protein function, structure, or stability, prior to hypothesizing a regulatory role. This aspect is critical and is not sufficiently addressed in the manuscript.

We thank the reviewer for raising this important point. In response, we conducted the following experiments: First, we measured the enzymatic activity of ADCY3 and found no significant differences between the T/T and C/C variant (Fig. EV1A). We then attempted to purify protein *in vitro* to further analyze its enzymatic kinetics. Unfortunately, despite optimizing various purification conditions, the purified protein tended to aggregate, and we were unable to obtain ADCY3 protein (ADCY3-Flag MW: 130.3 KDa) samples with satisfactory activity (See the figure below). To further evaluate the impact of this variant on protein structure, we performed protein structure prediction using AlphaFold Server. The root-mean-square-deviation (RMSD) was 1.3678 (Fig. EV1B), indicating that the C/C variant protein structure exhibits minimal differences from the T/T structure and maintains good stability. Based on these results, we propose that rs11676272 variant does not affect ADCY3 enzymatic activity.

Fig. 1 Flag-tag affinity chromatography and size exclusion chromatography (SEC) purification. **A–B.** SDS-PAGE from the the first Flag-tag affinity chromatography (A) and SEC profile following affinity chromatography purification (B). **C–D.** SDS-PAGE from the second Flag-tag affinity chromatography (C) and SEC profile following affinity chromatography purification (D). Aggregation protein is represented by ①, and degradation protein is represented by ② in C and D.

4. The observed difference in luciferase activity shown in Figure 1 (approximately 1.4-fold increase in T/T compared to the reference allele in HEK cells) appears modest, raising questions about its biological significance and how such an effect could translate into phenotypic consequences in mouse models.

We are grateful to the reviewer for raising this important question. We propose that this modest change in luciferase activity is sufficient to elicit significant in vivo phenotypic outcomes, based on two considerations: First, in vitro cellular models differ fundamentally from living organisms in terms of transcriptional microenvironment, epigenetic status, and cell-type specificity. Simple cell-based systems cannot fully recapitulate the complex in vivo physiological conditions, particularly in specialized contexts like the hypothalamus, where cell-type-specific

regulatory networks may amplify such subtle effect. Second, this phenomenon aligns with precedents in functional genomics. GWAS studies have shown that disease-associated variants often confer modest changes in reporter activity (1.2- to 1.5-fold) (Hua *et al*, 2018), yet these subtle regulatory alterations are sufficient to drive significant physiological or behavioral phenotypes during prolonged processes such as development or metabolic homeostasis.

5. The authors state that "it is not possible to manipulate ADCY3 in humans, resulting in a paucity of experimental evidence that could substantiate direct causality". This is incorrect: several human brain cell lines and induced pluripotent stem cells (iPSCs) differentiated into hypothalamic or neuronal lineages are currently available and could provide more relevant systems to study ADCY3 function in the context of obesity.

We agree with the reviewer's perspective and have corrected this inaccurate statement. The revised explanation is provided in lines 228–231 of the revised manuscript.

Minor comments:

The western blots shown in Figures 2 and 3 are not clean and lack sufficient resolution and contrast, which compromises the interpretation of the protein expression results.

We thank the reviewer for this constructive comment. We have repeated the Western blot experiments in Fig. 2 and 3 and have replaced the images in the revised manuscript with higher-quality blots that offer enhanced resolution and contrast for a clearer interpretation.

Referee #3:

Type 3 adenylyl cyclase (encoded by ADCY3) is predominantly expressed in neuronal primary cilia throughout the brain and a key enzyme mediating the cAMP signaling in the ciliary microdomain. ADCY3 is strongly associated with obesity, supported by numerous lines of GWAS studies and experimental studies on mouse models. The submitted report studied the causal relationship of SNP rs11676272 (T >

C) with obesity. It provides evidence showing that the rs11676272 SNP region very likely function as an eExon that preferentially binds to the transcription factor E2F3 to regulate AC3 expression, with the C risk allele decreasing the binding. There are multiple pieces of interesting data. The manuscript is well-written and figure panels nicely presented. The findings derived from molecular genetics study are solid and novel. The study also attempted to link down-regulated AC3 expression to ciliary function in the brain centers that regulate energy balance, claiming the reduced AC3 expression causes HFD-induce weight-gain by shortening primary cilia in the VMH. This claim is not well substantiated by presented data.

Major points:

(1) The mechanistic evidence on obesity is not compelling, as the cilia shortening effect observed in the VMH of the C allele knock-in mice is weak.

We thank the reviewer for this insightful comment. We have addressed this issue in the Discussion section as follows: “Although we hypothesize that ciliary shortening is an important factor contributing to obesity, we cannot exclude the involvement of additional mechanisms. Previous studies have shown that ADCY3 plays a crucial role in BAT, and the loss of its variant ADCY3-AT enhances energy expenditure and protects against obesity and metabolic disorders(Khani *et al.*, 2024). Moreover, ADCY3 is the predominant adenylate cyclase isoform expressed in WAT, and its mutation markedly reduces adenylate cyclase activity in this tissue(Pitman *et al.*, 2014). Our data also revealed pronounced hepatic steatosis in Mut-hADCY3 mice. It has been reported that ADCY3 activity is reduced in the liver of OB/OB mice, and liraglutide treatment significantly upregulates hepatic ADCY3 expression in obese mice, suggesting a potential role of hepatic ADCY3 in modulating the obesity phenotype(Liang *et al.*, 2016); In addition, GPR75-deficient mice are resistant to HFD-induced obesity, primarily attributed to alterations in feeding behavior; however, the authors also noted that cilia-related signaling may contribute to dietary responses(Jiang *et al.*, 2024). Similarly, we observed increased food intake in Mut-hADCY3 mice. Thus, while the rs11676272 variant promotes obesity in part through ciliary dysfunction, the altered feeding behavior and impaired ADCY3 signaling in

adipose tissue cannot be ruled out.”

(2) The markedly reduced AC3 protein expression in the mut-hADCY3 hypothalamus compared to that of hADCY3 does not align with the relatively mild shortening of primary cilia.

We appreciate the reviewer for raising this point. Our data show that while ADCY3 protein expression was significantly downregulated, the resulting ciliary shortening was modest. This discrepancy likely attributes to the non-linear relationship between ADCY3 expression and ciliary integrity. Notably, the ADCY3 levels we measured represent a tissue-wide average across the hypothalamus, whereas ciliary length is regulated by a finely tuned network of multiple genes—including MC4R, SSTR3, and IFT88—beyond ADCY3 alone. Thus, reduced overall ADCY3 expression may be partially buffered by compensatory mechanisms, or residual ADCY3 activity in specific neuronal subsets could be sufficient to maintain near-normal ciliary architecture. Together, these factors may account for the relatively mild ciliary phenotype observed at the macroscopic level.

(3) The effects of C allele knock-in on locomotor activity, energy expenditure, and overall neuronal activity were not assessed.

We thank and agree with the reviewer for this suggestion. The data on the effects of C allele knock-in on locomotor activity, energy expenditure, and overall neuronal activity have now been incorporated into Fig. 5F–H and Fig. EV6A–C of the revised manuscript.

(4) The SNP rs11676272 C risk allele likely affects either AC3 activity or expression systemically. A major limitation of the study is its exclusive focus on the VMH. The link between reduced cilia length in the VMH and HFD-induced weight gain is weak. Alternative mechanisms should be explored.

We thank the reviewer for this insightful comment. We fully agree that while the rs11676272 variant-induced reduction in ciliary length contributes significantly to the

obesity phenotype, additional mechanisms cannot be ruled out. For instance, altered ADCY3 expression may directly impair brown adipose tissue (BAT) and white adipose tissue (WAT) function, or indirectly disrupt energy metabolism through the regulation of food intake and related signaling pathways. These possibilities have been explicitly addressed in the Discussion section of our revised manuscript.

(5) The claim of a causal relationship, particularly regarding to obesity mechanism, appears to be an over-interpretation, as the underlying mechanisms of AC3 association with obesity remain unclear and ciliary defects-caused obesity can result from multiple contributing factors.

We appreciate the reviewer's insightful observation. We acknowledge that the statement regarding a causal relationship was too strong, as our study does not provide definitive evidence for the exact mechanism. To clarify, our intention was not to assert a definitive mechanism but to propose a testable hypothesis based on the known biology of ADCY3 in ciliary function. As elaborated in the Discussion, we suggest that the association between the rs11676272 variant and obesity *could be mediated* through a potential influence on ciliary length. We fully acknowledge that the underlying mechanisms are multifactorial and not yet fully resolved, and we have revised the text to ensure our language more carefully presents this as a plausible model for future validation.

Minor points:

(1) Abstract: "providing offering", please delete one.

We thank the reviewer and apologize for the confusion. We have removed the repeated text in the revised manuscript.

(2) Statistical analysis: A normality test should be performed to confirm that the data are normally distributed before applying Student's t-test. If normality cannot be assumed, the Mann-Whitney U test should be used for unpaired samples, and the Wilcoxon signed-rank test for paired samples.

We thank the reviewer for this suggestion. We have now performed normality tests on the data. In accordance with the recommendation, non-normally distributed data were analyzed using the Mann-Whitney U test for unpaired samples and the Wilcoxon signed-rank test for paired samples. The statistical analysis section has been revised accordingly in the revised manuscript.

References

- Hua JT, Ahmed M, Guo H, Zhang Y, Chen S, Soares F, Lu J, Zhou S, Wang M, Li H *et al* (2018) Risk SNP-Mediated Promoter-Enhancer Switching Drives Prostate Cancer through lncRNA PCAT19. *Cell* 174: 564-575.e518
- Jiang Y, Xun Y, Zhang Z (2024) Central regulation of feeding and body weight by ciliary GPR75. *The Journal of clinical investigation* 134: e182121
- Kang Y, Kang X, Yang H, Liu H, Yang X, Liu Q, Tian H, Xue Y, Ren P, Kuang X *et al* (2022) *Lactobacillus acidophilus* ameliorates obesity in mice through modulation of gut microbiota dysbiosis and intestinal permeability. *Pharmacological research* 175: 106020
- Khani S, Topel H, Kardinal R, Tavanez AR, Josephrajan A, Larsen BDM, Gaudry MJ, Leyendecker P, Egedal NM, Güller AS *et al* (2024) Cold-induced expression of a truncated adenylyl cyclase 3 acts as rheostat to brown fat function. *Nature metabolism* 6: 1053-1075
- Liang Y, Li Z, Liang S, Li Y, Yang L, Lu M, Gu HF, Xia N (2016) Hepatic adenylate cyclase 3 is upregulated by Liraglutide and subsequently plays a protective role in insulin resistance and obesity. *Nutrition & diabetes* 6: e191
- Oya M, Miyasaka Y, Nakamura Y, Tanaka M, Suganami T, Mashimo T, Nakamura K (2024) Age-related ciliopathy: Obesogenic shortening of melanocortin-4 receptor-bearing neuronal primary cilia. *Cell metabolism* 36: 1044-1058.e1010
- Pitman JL, Wheeler MC, Lloyd DJ, Walker JR, Glynne RJ, Gekakis N (2014) A gain-of-function mutation in adenylate cyclase 3 protects mice from diet-induced obesity. *PloS one* 9: e110226
- Yang D, Wu X, Wang W, Zhou Y, Wang Z (2022) Ciliary Type III Adenylyl Cyclase in the VMH Is Crucial for High-Fat Diet-Induced Obesity Mediated by Autophagy. *Advanced science (Weinheim, Baden-Wurttemberg, Germany)* 9: e2102568
- Yu X, Benitez G, Wei PT, Krylova SV, Song Z, Liu L, Zhang M, Xiaoli AM, Wei H, Chen F *et al* (2024) Involution of brown adipose tissue through a Syntaxin 4 dependent pyroptosis pathway. *Nature communications* 15: 2856

Dear Dr. Hu,

Thank you for the submission of your revised manuscript. We received the enclosed reports from the referees as well as cross-comments. As you will see, all referees have some more suggestions for how the study should be further strengthened, and I think all points should be addressed, except that your explanation for why you use HEK cells is fine and this comment does not need to be addressed again. Please co-submit a detailed point-by-point response with your newly revised ms.

There are also a few editorial requests that will need to be addressed:

- The "Data and materials availability" section needs to be deleted. Instead, please add a "Data Availability Section" (DAS) that should list all data generated in this study and deposited in public databases, including the specific URLs. If no such data have been generated, please add this explanation to the DAS.
- The conflict of interest subheading needs to be renamed to Disclosure and Competing Interests Statement.
- The FUNDING INFO is missing in the ms, it needs to be part of the Acknowledgments section. One funder needs to be added to our online submission system: Hebei University (050001-5000019). All funding info needs to be in the ms file and in our online submission system.
- Figure CALLOUTS are missing in the ms file for the following figures: Fig. 2B, Fig. 3PQ, Fig. 7I, please add.
- Table EV1 and Table EV2 look more like Datasets and should be renamed to Dataset EV1 and Dataset EV2, and the remaining two tables should then be Table EV1 and Table EV2; this needs to be corrected in all places (source file names, titles in the system, legends, callouts in the ms including Reagents Table) and the term "supplementary" should not be used. All tables need to have a title within the table file and the EV tables should only have 1 tab (Datasets can have more tabs).
- There is also a Supplementary Data zip folder, but I cannot open the files. Can you please explain what this is and if it needs to be published? It could may be be another Dataset.
- Please delete all instructions and examples from the Reagents & Tools Table file.
- For the Source Data, please upload one Source Data folder per main figure.
- The manuscript sections should be in the following order: Title page - Abstract & Keywords - Introduction - Results - Discussion - Methods - Data Availability - Acknowledgments - Disclosure Statement & Competing Interests - References - Figure Legends - (Main Tables with legends if applicable) - Expanded View Figure Legends.
- Supporting Information section needs to be removed from the ms
- Our routine image analyses found two issues:
 1. Figure 3P top line ADCY3 contains a pixelated smudge. This can also be seen in the source data. Can you please provide the original uncropped blot from the moment of capture.
 2. Figure EV3B contains an undisclosed splice site. Please address this by adding a clear markaton line highlighting the splice. Please submit source data for this figure to verify the integrity of the figure.

* Figure Legends - Comments *

- Please note that the legend for figure 3P, Q is missing in the manuscript. This needs to be rectified.
- Please note that the exact p values are not provided in the legends of figures 1F, H, I; 2D, E, G, H, J; 3C, E, F, H, I, K, L, M, O, Q; 4C, 5C, D, J; 6A, C, F, J; 7A, C, I; EV2 A, B; EV4 A, EV7 A, E. Please provide exact p-values as reasonable.
- Please note that information related to n is missing in the legend of figure 3Q

I would like to suggest some minor changes to the abstract that needs to be written in present tense. Please let me know whether you agree with the following:

Genome-wide association studies (GWASs) have identified hundreds of obesity-associated SNPs, but establishing their causality remains challenging. Here, we demonstrate that rs11676272, located in the ADCY3 gene, is a functional causal variant for obesity susceptibility. Bioinformatic analyses and dual-luciferase reporter assays indicate that the rs11676272 region may act as a human-gained enhancer regulating ADCY3 expression. In HEK293T cells, CRISPR-Cas9-mediated single-nucleotide editing of rs11676272 (T > C) reduces ADCY3 expression. Moreover, the rs11676272-T allele is preferentially bound by the

transcription factor E2F3 to upregulate ADCY3 expression, whereas the rs11676272-C risk allele loses this binding. In vivo, the rs11676272 T > C variant in human ADCY3 (hADCY3) knock-in mice accelerates weight gain under high-fat diet conditions and shortens primary cilia in the ventromedial hypothalamus (VMH). CRISPRa-mediated activation of the hADCY3 promoter region rescues ciliary length in both the VMH and hypothalamic arcuate nucleus of Mut-hADCY3 mice. Our data reveal a causal role for rs11676272 in obesity, offering insight into potential therapeutic strategies.

EMBO press papers are accompanied online by A) a short (1-2 sentences) summary of the findings and their significance, B) 2-3 bullet points highlighting key results and C) a synopsis image that is exactly 550 pixels wide and 200-600 pixels high (the height is variable). The synopsis image should provide a sketch of the major findings, like a graphical abstract. Please note that text needs to be readable at the final size. Please send us this information along with the final manuscript.

I look forward to seeing a newly revised version of your manuscript as soon as possible.

Referee #1:

The authors attempted to address the concerns brought up by the reviewers.

Referee #2:

The authors did not adequately address my question regarding the fine-mapping, even though its feasibility is straightforward in this context. As a result, causality cannot be inferred at this stage. Direct functional assays are insufficient to establish causality when multiple SNPs are in LD. The manuscript does not discuss the potential contribution of other variants in strong LD, which is essential for interpreting the association signal.

The current in vitro analyses do not provide convincing evidence, particularly given the absence of detectable enzymatic activity, despite the fact that the protein is fundamentally an enzyme.

The authors also did not succeed in performing in-depth in vitro assays in relevant cellular models, which represents a major limitation. HEK293T cells are kidney-derived and bear no biological relevance to the hypothalamus or to the physiological context in which the gene is normally expressed. This strongly limits the interpretability of the functional results.

In addition, the authors were unable to examine the functional impact of the missense variant. The reliance on AlphaFold predictions is insufficient, as AlphaFold is known to produce false-negative results and has limited sensitivity for assessing subtle structural or functional disruptions caused by missense variants.

Referee #3:

The revisions have adequately addressed the reviewers' concerns and incorporated new data. The manuscript is now suitable for publication in EMBO Reports.

Cross-comments from referee 1:

I understand the authors attempted to respond. However, I was not impressed on adenylyl cyclase activity measurement comparing the WT and mutant. There were not unit indications in the graph and they could not measure activity after purification of the WT and mutant. I am not that enthusiastic overall.

Cross-comments from referee 3:

I am not comfortable with the strength of the conclusions, for example in the Abstract, which states, "Our data reveal the causal mechanism of rs11676272 in obesity." This claim is too strong. The mechanisms by which defects in AC3 or primary cilia lead to obesity are not yet established.

Responses to Reviewers

We thank the three reviewers for their critical comments. Our replies are detailed under each specific comment below in blue, and we also labeled our modifications in red in the revised manuscript.

Referee 1:

The authors attempted to address the concerns brought up by the reviewers.

Referee 2:

The authors did not adequately address my question regarding the fine-mapping, even though its feasibility is straight forward in this context. As a result, causality cannot be inferred at this stage. Direct functional assays are insufficient to establish causality when multiple SNPs are in LD. The manuscript does not discuss the potential contribution of other variants in strong LD, which is essential for interpreting the association signal.

We thank the reviewer for the important comment regarding fine-mapping analysis. We have now performed comprehensive linkage disequilibrium (LD) and statistical fine-mapping analyses to evaluate potential contributions of other variants in strong LD with rs11676272. The relevant content has been added to the Results section as follows: To assess whether the BMI association at the *ADCY3* locus is driven by rs11676272 independently of other linked variants, we analyzed pairwise LD among 123 SNPs using LDlink (Machiela & Chanock, 2015), with the European population from the 1000 Genomes Project Phase 3 as the reference panel. Statistical fine-mapping was performed using the approximate Bayes factor (ABF) framework (Wakefield, 2009) to estimate posterior inclusion probabilities (PIP), assuming a prior variance of 0.04. The 95% credible set was defined as the minimal set of variants with a cumulative PIP exceeding 0.95 (Table EV2). Fine-mapping provided strong evidence that rs11676272 is the causal variant underlying the BMI association at the *ADCY3* locus. This variant showed a PIP greater than 0.999, and the 95% credible set contained rs11676272 alone (Fig. 1C, Table EV2). Among the 123 variants analyzed, 120 were in LD with

rs11676272, including 20 high-LD SNPs ($r^2 > 0.8$), 39 moderate-LD SNPs ($0.5 \leq r^2 < 0.8$), 30 lower-LD SNPs ($0.2 \leq r^2 < 0.5$), and 31 low-LD SNPs ($r^2 < 0.2$). Association strength declined monotonically with decreasing LD, with mean $-\log_{10}P$ values of 76.7 ($r^2 = 0.8-1.0$), 72.2 ($r^2 = 0.6-0.8$), 35.3 ($r^2 = 0.4-0.6$), and 8.8 ($r^2 < 0.2$). This pattern is consistent with an LD-driven signal decay rather than multiple independent effects. Although 14 low-LD SNPs ($r^2 < 0.2$) reached genome-wide significance ($P < 5 \times 10^{-8}$), their association signals were 10–15 orders of magnitude weaker than that of rs11676272 (Table EV2), suggesting that they reflect long-range LD or haplotypic effects rather than independent causal variants.

While LD limits causal inference from association data alone, the convergent evidence from fine-mapping ($P = 5.01 \times 10^{-88}$, PIP > 0.999, single-variant 95% credible set), functional annotation (exonic location, eQTL evidence, RegulomeDB score of 1b), and experimental validation supports rs11676272 as the causal variant driving the BMI association at the *ADCY3* locus.

In addition, the authors were unable to examine the functional impact of the missense variant. The reliance on AlphaFold predictions is insufficient, as AlphaFold is known to produce false-negative results and has limited sensitivity for assessing subtle structural or functional disruptions caused by missense variants.

We appreciate the reviewer's valid point regarding the functional assessment of the missense variant. We agree that AlphaFold predictions have limitations in detecting subtle structural or functional perturbations caused by missense variants and may yield false negatives. Owing to persistent difficulties in purifying the *ADCY3* protein, we cannot entirely exclude an effect of rs11676272 variant on enzymatic activity, despite the lack of detectable alterations in AlphaFold structural modeling or enzymatic assays. Nevertheless, consistent evidence from cellular and in vivo experiments clearly establishes that rs11676272 resides in an enhancer region and modulates *ADCY3* expression by modulating enhancer activity. Accordingly, the central conclusion of our study—that rs11676272 regulates *ADCY3* expression through an enhancer-mediated mechanism—remains firmly supported, irrespective of its potential effects on

enzymatic activity. We have incorporated a discussion of these points and limitations into the revised manuscript to enhance the rigor of our interpretations.

Referee 3:

The revisions have adequately addressed the reviewers' concerns and incorporated new data. The manuscript is now suitable for publication in EMBO Reports

Cross-comments from referee 1:

I understand the authors attempted to respond. However, I was not impressed on adenylyl cyclase activity measurement comparing the WT and mutant. There were not unit indications in the graph and they could not measure activity after purification of the WT and mutant. I am not that enthusiastic overall.

I sincerely apologize for any confusion caused. To address the reviewer's concern, we have now clearly indicated the units in the revised Fig. EV1A. Regarding the measurement of adenylyl cyclase activity using purified proteins, we encountered significant technical challenges in obtaining sufficient quantities of correctly folded and active recombinant ADCY3 protein for both the wild-type and mutant variants. We have now discussed this limitation more explicitly in the revised Discussion section to clarify the experimental constraints.

Cross-comments from referee 3:

I am not comfortable with the strength of the conclusions, for example in the Abstract, which states, "Our data reveal the causal mechanism of rs11676272 in obesity." This claim is too strong. The mechanisms by which defects in AC3 or primary cilia lead to obesity are not yet established.

Thank you for raising this important point regarding the strength of our conclusions. We agree that the original statement, "Our data reveal the causal mechanism of rs11676272 in obesity" may overstate the certainty of our findings. To address this point, we have revised the sentence in the Abstract (line 21) to more accurately reflect that our evidence is strongly supportive but not definitive.

- Machiela MJ, Chanock SJ (2015) LDlink: a web-based application for exploring population-specific haplotype structure and linking correlated alleles of possible functional variants. *Bioinformatics (Oxford, England)* 31: 3555-3557
- Wakefield J (2009) Bayes factors for genome-wide association studies: comparison with P-values. *Genetic epidemiology* 33: 79-86

Dr. Xiaoyu Hu
Hebei University
School of Life Sciences
180 Wusi East Road, Baoding, Hebei Province
Baoding, Hebei Province 071002
China

Dear Dr. Hu,

I am very pleased to accept your manuscript for publication in the next available issue of EMBO reports. Thank you for your contribution to our journal.

You may qualify for financial assistance for your publication charges - either via a Springer Nature fully open access agreement or an EMBO initiative. Check your eligibility: <https://link.springer.com/journal/44319/how-to-publish-with-us>

Referee #2:

The authors have now adequately addressed my comments.

>>> Please note that it is EMBO Reports policy for the transcript of the editorial process (containing referee reports and your response letter) to be published as an online supplement to each paper. If you do NOT want this, you will need to inform the Editorial Office via email immediately. More information is available here: <https://link.springer.com/partners/embo-press/editorial-policies#Peer%20review>